# ROBOTIC MANIPULATION BY IMITATING GENERATED VIDEOS WITHOUT PHYSICAL DEMONSTRATIONS

**Shivansh Patel**[1]    **Shraddhaa Mohan**[1]    **Hanlin Mai**[1]
**Unnat Jain**[2*]    **Svetlana Lazebnik**[1*]    **Yunzhu Li**[3*]
[1]UIUC    [2]UC Irvine    [3]Columbia University

## ABSTRACT

This work introduces Robots Imitating Generated Videos (RIGVid), a system that enables robots to perform complex manipulation tasks—such as pouring, wiping, and mixing—purely by imitating AI-generated videos, without requiring any physical demonstrations or robot-specific training. Given a language command and an initial scene image, a video diffusion model generates potential demonstration videos, and a vision-language model (VLM) automatically filters out results that do not follow the command. A 6D pose tracker then extracts object trajectories from the video, and the trajectories are retargeted to the robot in an embodiment-agnostic fashion. Through extensive real-world evaluations, we show that filtered generated videos are as effective as real demonstrations, and that performance improves with generation quality. We also show that relying on generated videos outperforms more compact alternatives such as keypoint prediction using VLMs, and that strong 6D pose tracking outperforms other ways to extract trajectories, such as dense feature point tracking. These findings suggest that videos produced by a state-of-the-art off-the-shelf model can offer an effective source of supervision for robotic manipulation. Project page: https://rigvid-robot.github.io/

## 1 INTRODUCTION

Videos offer a rich and expressive source of training data for robotic manipulation, and numerous methods have successfully leveraged them for supervision. Such methods typically fall into two categories: (1) Learning from publicly available large-scale datasets of real-world videos (Bahl et al., 2023; Ye et al., 2024), and (2) Imitation of specific demonstrations captured under controlled conditions that closely match the execution setting (Bahl et al., 2022; Kareer et al., 2024). Unfortunately, both of these strategies come with challenges that limit broad deployment. Large-scale video datasets often introduce domain gaps (Zhou et al., 2024; Gao et al., 2025) and require adaptation to specific robot embodiments and tasks (Bahl et al., 2023; O'Neill et al., 2023). On the other hand, video-based imitation involves laborious data collection that must ensure close alignment in viewpoints, morphologies, and interaction modalities (Bahl et al., 2022; Dasari & Gupta, 2021).

Motivated by recent advances in video generation, we explore a new paradigm: can a **single generated video**, generated to exactly match our input environment and task description, be used as the sole source of supervision for robotic manipulation? Recently released models like SORA (Brooks et al., 2024) and Kling (Kli, 2024) have demonstrated impressive capabilities in producing realistic videos from language and image inputs. At the same time, it has been shown that such videos can suffer from distorted object geometries (Liu et al., 2024; Zhang et al., 2024), physically implausible interactions (Motamed et al., 2025; Yang et al., 2025), and unrealistic scene dynamics (Bansal et al., 2024; Guo et al., 2025). Consequently, while the idea of synthesizing video demonstrations is enticing, its usefulness in the robotics setting is yet to be convincingly established. Prior work incorporating video generation into robotics typically relies on additional supervision, such as task-specific training (Du et al., 2024) or fine-tuning on offline robot trajectory datasets (Bharadhwaj et al., 2024b;a). By contrast, we ask whether a robot can perform real-world manipulation tasks solely by imitating generated videos—*without any additional supervision or task-specific training.*

---

[*]denotes equal advising.

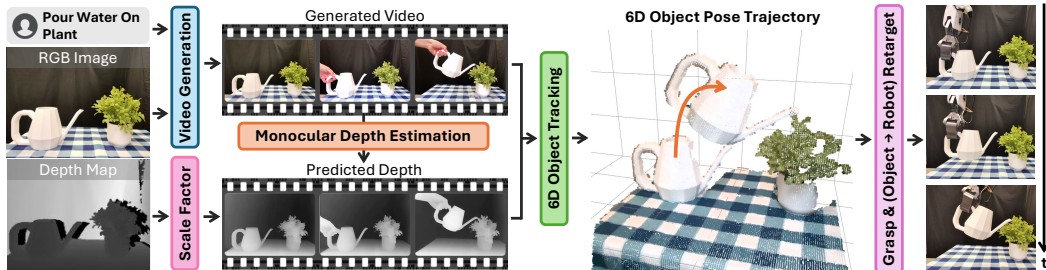

Figure 1: **RIGVid overview.** Given an initial scene image and depth, we generate a video conditioned on a language command. A VLM-based automatic filtering step (not shown) can be used to reject videos that fail to follow the prompt. A monocular depth estimator recovers depth for each frame of the generated video, and these depth maps are combined with the corresponding RGB frames to produce 6D Object Pose Trajectory. After grasping, the trajectory is retargeted to the robot for execution.

To this end, we introduce **Robots Imitating Generated Videos (RIGVid)**, a framework that connects video generation models to real-world robotic execution. Fig. 1 gives an oveview of the method. Given an input RGB-D image of the scene and a free-form language command (e.g., "pour water on the plant"), we use a state-of-the-art video diffusion model to generate a candidate video of the task. The generated video is not guaranteed to accurately follow the language command – but we show that a VLM can be used to automatically filter out unsuccessful generations with high accuracy. Next, we estimate per-frame depth on the video, segment the manipulated object, and track its *6D object pose trajectory* across the frames using the FoundationPose tracker (Wen et al., 2023b). While this tracker relies on a pre-computed object mesh, preliminary experiments (App. C) indicate that our method is compatible with mesh-free approaches, though their inference speed is currently infeasible for real-time deployment. Finally, the extracted 6D object pose trajectory is retargeted to the robot for execution. The retargeting process only requires the transformation between the end-effector and the object, so it can be easily applied across platforms. During deployment, RIGVid performs real-time object tracking and dynamically adjusts the robot's actions to handle disturbances and execution-time variations, promoting robust and adaptive behavior.

We evaluate RIGVid on four real-world manipulation tasks: pouring water, lifting a lid, placing a spatula on a pan, and sweeping trash. These tasks span diverse manipulation challenges, including a range of depth variation (minimal in pouring vs. significant in lifting), thin and partially occluded objects (spatula, sweeping brush), and diverse object geometries and actions. Our results show that, when paired with our filtering mechanism, generated videos are as effective as human videos for visual imitation, enabling robots to act entirely from synthetic supervision. Moreover, the performance of RIGVid improves with video quality, suggesting a favorable trend where advances in generative models directly translate to stronger manipulation capabilities.

The main downside of video generation is its substantial computational cost. Also, on a representational level, one may wonder whether predicting video pixels is wasteful, and whether we should instead predict a compact and minimal representation that can be efficiently translated to an executable trajectory. One example of this philosophy is the recent ReKep method (Huang et al., 2024a), which uses a VLM to generate relational keypoint constraints from a task description and then solves for a 6D trajectory. We compare our approach to ReKep and demonstrate that video generation does, in fact, perform substantially better than the generation of a more sparse and high-level representation. Next, given a generated video, one may ask whether 6D object-level tracking is necessary, given its up-front requirement of an object mesh. To address this question, we compare against a broad range of alternative tracking approaches — sparse point tracking (Bharadhwaj et al., 2024b), dense optical flow (Ko et al., 2023), 3D feature fields (Kerr et al., 2024), and generated goal supervision (Bharadhwaj et al., 2024a) — and show consistently higher success rates.

In summary, our contributions are: (1) We propose a framework that enables robots to perform open-world manipulation tasks without any real-world demonstrations, only generated videos. (2) We show high-quality generated videos perform on par with real videos for robotic imitation, establishing that synthetic data can serve as an effective substitute for real data. (3) We demonstrate that our combination of video generation and 6D trajectory extraction outperforms competing SOTA methods based on VLMs, point tracking, optical flow, feature fields, and generated-goal supervision.

## 2 RELATED WORK

**Direct Imitation from Videos.** This seeks to match visual states in demonstration videos to those of the learner, without requiring expert action labels or robot states (Dasari & Gupta, 2021; Valassakis et al., 2022). While effective, this approach demands paired demonstrations in the same setting. A common strategy is to leverage visual correspondences—tracks (Bharadhwaj et al., 2024b) or optical flow (Argus et al., 2020; Xu et al., 2024)—to infer object trajectories. Bharadhwaj *et al*. (Bharadhwaj et al., 2024b) predicts object tracks and uses PnP to recover poses for closed-loop task execution. Dense descriptor learning (Florence et al., 2018; Zhu et al., 2024) has proven powerful for handling variations in object geometry and appearance. Kerr *et al*. (Kerr et al., 2024) recover object part trajectories from monocular videos using feature fields. Crucially, these methods rely on demonstrations collected under conditions closely matching the target task. In contrast, our method removes this requirement by generating task and scene-conditioned videos.

**Imitation from Offline Videos.** This paradigm alleviates the need for paired demonstrations by leveraging offline video data, and has consequently attracted significant attention (Smith et al., 2019; Liu et al., 2018). Many works focus on learning affordance models from internet-scale video datasets (Bahl et al., 2023; Srirama et al., 2024). Here, affordances are defined as scene-centric predictions of where and how an agent can interact, often captured as contact-point heatmaps and short motion trajectories that can be translated into robot actions. For example, Bahl *et al*. (Bahl et al., 2023) learn from large-scale human videos to output dense contact maps and trajectory waypoints, which downstream imitation, exploration, or reinforcement modules can transform into executable robot motions. However, these methods suffer from domain gap between training videos and task-specific environments, and require additional mechanisms to obtain task-conditioned affordances. In contrast, our method does not explicitly predict affordances, but instead relies on generated, task- and scene-specific generated videos for imitation.

**Video Generation for Robotics.** Video generation has emerged as a promising avenue for robotics (Du et al., 2024; 2023). A common limitation of these is their reliance on robot data, either to train the video generation model (Liang et al., 2024; Sun et al., 2024), or to train policies (Bharadhwaj et al., 2024a), or both (Du et al., 2024; 2023). Bharadhwaj *et al*. (Bharadhwaj et al., 2024a) leverages tracks on generated videos to condition policy learning. Albaba *et al*. (Albaba et al., 2025) uses generated videos to compute rewards for RL training. The most closely related work is Liang *et al*. (Liang et al., 2024), which executes robotic tasks by tracking tools attached to the robot's end effector. While effective, their method relies on 1,822 human-collected robot demonstrations for four tasks, and is confined to tasks executable only by tools. In contrast, our approach requires no such data collection. Instead of tools, our method tracks objects, enabling a significantly broader range of manipulation tasks without using any robot data.

**Motion Retargeting for Object Manipulation.** Early work in learning from demonstration established the foundation for motion retargeting (Gleicher, 1998; Calinon, 2016). More recently, deep learning-based retargeting methods have emerged (Cheng et al., 2024; Choi et al., 2020), with some incorporating object-centric representations to bridge the gap between the demonstrator and the robot (Kerr et al., 2024; Li et al., 2024). Many approaches have applied retargeting to humanoid robots (Hu et al., 2014). Other works have extended these techniques to dexterous manipulation (Qin et al., 2022; Lakshmipathy et al., 2024). Like most prior work, we assume a fixed transformation between the end-effector and the object. While motion retargeting has traditionally relied on human demonstrations, RIGVid eliminates this dependency by leveraging generated videos.

## 3 OUR METHOD: ROBOTS IMITATING GENERATED VIDEOS

An overview of our method is shown in Fig. 1. It takes as inputs the initial scene RGB image, its corresponding depth map, and a free-form human command. Our goal is to predict the robot's 6DoF end-effector trajectory. This section describes the key steps of RIGVid: (1) Generate a scene and task-conditioned video and predict its corresponding depth using a monocular depth estimator (Sec. 3.1); (2) Compute 6D pose rollout via an object pose tracker (Sec. 3.2); (3) Grasp the object and retarget the pose trajectory to the robot, and execute the resulting trajectory (Sec. 3.3).

### 3.1 GENERATING VIDEOS AND CORRESPONDING DEPTH

Since the generated videos may not necessarily follow the language command or have other issues, we need an automatic filtering mechanism to discard inaccurate generations. We found that we can

do the filtering reliably by prompting a VLM – specifically, GPT-4o (Achiam et al., 2023) – to assess whether the generated video depicts a successful execution of the command. As image input to GPT-4o, we sample four evenly spaced frames in the video and concatenate them vertically to create a video summary. The VLM determines whether the action described in the command is performed by a visible hand. App. B provides the full prompt used for filtering and examples of video summaries with their corresponding VLM responses. If the response is negative, we regenerate the video and repeat the process for up to five attempts. If all attempts fail, we default to the final attempt.

As input to the downstream tracking step, we also need to predict the depth for the generated video, using the predictor from Ke *et al.* (Ke et al., 2024). One complication is that the estimated depth is not grounded in real-world units, but subject to a scale-shift ambiguity (Hartley & Zisserman, 2003). Consistent with prior works adopting depth estimators in vision-based robotics (Gervet et al., 2023), we compute an affine scale-and-shift transformation, aligning the predicted depth in the first frame with the initial real depth map around the active object (discussed in Sec. 3.2). This transformation is then applied to the entire predicted video to resolve the ambiguity.

## 3.2 IDENTIFYING ACTIVE OBJECT MASK AND 6D OBJECT POSE TRAJECTORY

To extract 6D pose rollout, we first identify the active object—the one being manipulated in the generated video. A binary mask for this object in the initial RGB image is essential for object tracking and determining which object to grasp. Given the initial image and the task command, we prompt GPT-4o to identify the object most likely to be manipulated. We then ground the predicted object category into a bounding box using Grounding DINO (Liu et al., 2023), and further refine this into a segmentation mask using SAM-2 (Ravi et al., 2024).

Once the active object is localized by the mask, we track it across the generated video using the scaled predicted depth. This yields the 6D pose rollout. Tracking objects in videos is a rich area of research, and we experimented with several 6D pose trackers (Labbé et al., 2022; Wen et al., 2023a;b). For real-world deployment, we found FoundationPose (Wen et al., 2023b) to perform the best. It requires an object mesh, which we pre-compute using BundleSDF (Wen et al., 2023a). For this, we record a short RGBD video where the object is held and rotated in front of the camera to capture all sides. While straightforward, this process constrains our method to settings where a mesh can be precomputed. Nonetheless, as shown in App. C, our method is also compatible with mesh-free approaches—BundleSDF can jointly reconstruct and track the object—but current inference speeds make these alternatives infeasible for real-time use. To ensure stable and realistic motion during execution, we apply an averaging filter to smooth abrupt pose changes, particularly in rotation. Additional details on this smoothing step are provided in App. D.

## 3.3 OBJECT TO ROBOT MOTION RETARGETING

Once the object trajectory is obtained, we first grasp the object. We use an off-the-shelf grasper, AnyGrasp (Fang et al., 2023), to identify and execute the highest-scoring grasp within a defined boundary around the active object mask. After grasping, we retarget its trajectory to the robot's end-effector. Since the object remains firmly grasped, we assume a fixed transformation between the robot's end-effector and the object. This transformation is obtained by composing two rigid-body transforms: (1) the pose of the object relative to the gripper at the moment it is grasped and (2) the offset between the gripper and the robot's end-effector. By combining these two components, we obtain a single transformation from the end-effector to the object.

The corresponding end-effector trajectory is obtained by applying the fixed end-effector-to-object transformation to the object's pose along the entire trajectory. This formulation ensures that the retargeted 6D pose rollout follows the object's motion while maintaining a stable grasp. These are executed on the physical robot, enabling it to reproduce the object's movement as observed in the generated video. A key strength of this approach is that it is robot-agnostic. Specifically, to accommodate a different robot or gripper, only the end-effector to the object transformation needs to be updated to reflect the new end-effector configuration.

## 3.4 CLOSED LOOP EXECUTION

A core strength of our approach is its ability to operate in a closed-loop manner, enabling robust execution despite disturbances or unexpected changes during task execution. During deployment, the system continuously tracks the object's 6D pose in real time using FoundationPose to update the robot's end-effector trajectory as the task progresses. This feedback allows the robot to dynamically

adjust its motions: if the object deviates from the planned trajectory due to external perturbations, such as a human pushing the robot or a slip after grasping, the system detects the discrepancy by comparing the current object pose to the precomputed trajectory. If the detected deviation exceeds a threshold of 3 cm in position or 20 degrees in orientation, the robot backtracks to the last successfully executed trajectory point and resumes execution from there (Fig. 2). This recovery mechanism enables RIGVid to maintain stable task execution, realigning and successfully completing the manipulation task despite perturbations. Additional examples of robustness are provided in App. H.

## 4 EXPERIMENTS

This section presents our experimental evaluation. We describe the robot setup, manipulation tasks, and evaluation protocol (Sec. 4.1). Then assess the impact of video generation models and filtering strategies on downstream robotic performance (Sec. 4.2). Next, we compare RIGVid to SOTA VLM-based trajectory prediction method that allows zero-shot execution (Sec. 4.3), and to alternative tracking approaches for trajectory extraction (Sec. 4.4). Finally, we test generalization across embodiments, extensions to new tasks, and robustness to real-world disturbances (Sec. 4.5).

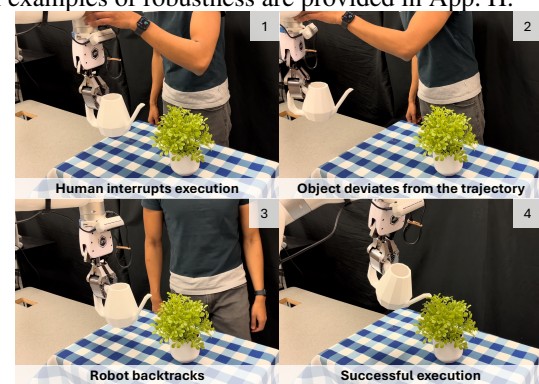

Figure 2: **RIGVid is robust to perturbations.** A human pushes the robot during execution (image 1), causing the object to deviate from the trajectory. When the deviation is detected (image 2), the robot backtracks to the last successfully executed trajectory point (image 3) and then resumes the planned motion (image 4).

### 4.1 ROBOT
### SETUP, TASKS, AND EVALUATION

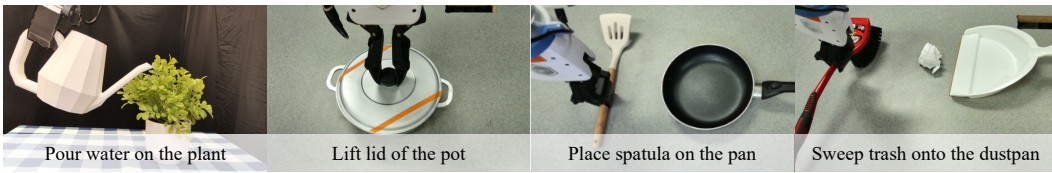

Figure 3: **Evaluation tasks.** We evaluate RIGVid on everyday manipulation tasks of varying difficulty.

We conduct experiments on an xArm7 robot arm with a stationary Orbbec Femto Bolt camera, positioned next to the robot to capture RGBD observations. We evaluate our method on four everyday manipulation tasks, which are illustrated in Fig. 3. These span a diverse range of robotic challenges, and their descriptions are as follows:

1. **Pouring water** requires the robot to position and tilt a watering can above a plant. While the depth of the can relative to the camera remains largely constant, successful execution demands a smooth trajectory spanning the pick-up, transport, and pouring phases. A trial is considered successful if the watering can's spout is positioned above the plant at the end of the execution.

2. **Lifting a lid** requires the robot to lift a pot lid. Unlike pouring, where the camera is viewing the scene from the side, the camera here is looking down towards the pot. As a result, this task involves significant variation in object depth, as the lid moves closer to the camera during execution. Success is achieved if the lid is no longer in contact with the pot at the end of the trial.

3. **Placing a spatula on a pan** requires the robot to place the head of a spatula into a pan. The spatula has a thin, elongated geometry and is often partially occluded during manipulation, which presents a challenge for object tracking, particularly for non-mesh-based approaches. The task is considered successful if the spatula's head is in the pan at the end of execution.

4. **Sweeping trash** requires the robot to sweep trash into a dustpan. This task is especially challenging as it combines the need for precise positioning to align the head of the sweeping brush with the trash, along with all the challenges encountered from the placing task. A trial is successful if the trash is touching the base of the dustpan at the end of the execution.

Task success is determined via human judgment based on the above criteria, though the procedure could be readily automated with a VLM. The initial setup configuration is fixed across trials of the

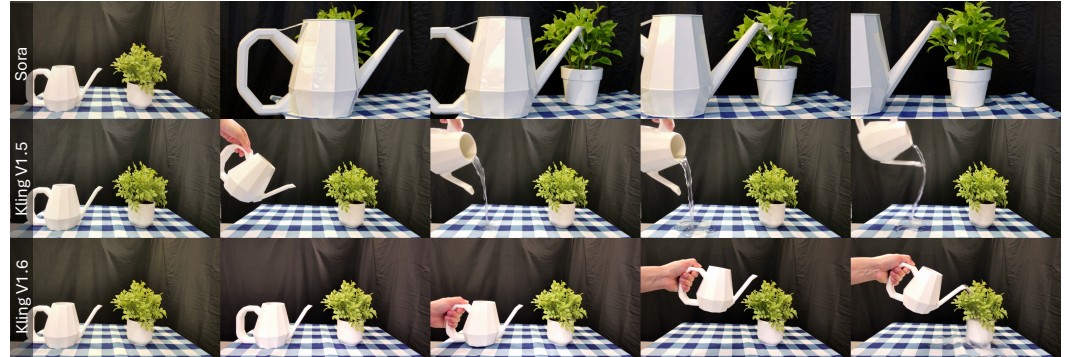

Figure 4: **Qualitative comparison of video generation for three models.** Sora (top) drastically alters the scene layout and object size. Kling v1.5 (middle) does not fully follow the prompt (water not poured over the plant) and exhibits physically implausible behaviors (water pouring out of the top of the kettle but not the spout). Kling v1.6 (bottom) produces the most consistent and realistic result.

same task, and each trial has a different generated video. All baselines use the same videos for consistent comparison and reporting.

## 4.2 QUALITY AND FILTERING OF GENERATED VIDEOS

As discussed in Sec. 3.1, we experimented with Sora, Kling v1.5, and Kling v1.6 for video generation. We also compared different video filtering strategies. Next, we present our key findings.

*How do different video generation models compare for robotic imitation?* Sora is known for creating visually impressive and cinematic videos. Unfortunately, these videos often prioritize aesthetics over following the human command. As seen in the top row of Fig. 4, Sora frequently alters the camera viewpoint, changes object positions, or even swaps out objects mid-sequence. This lack of scene and object consistency makes Sora poorly suited for imitation. Kling v1.5 places more emphasis on following language instructions, generally preserves the original scene, and correctly depicts the target object. Nonetheless, it is still prone to physically implausible behaviors and command following failures. In the second row of Fig. 4, the teapot is not positioned over the plant and the water pours out from the top, not the spout (in other failure cases, nothing at all happens in the video, and the command is not even attempted). By contrast, Kling v1.6 (bottom row of Fig. 4) has greatly improved command following and physical plausibility, proving to be the most reliable video generator for us. More examples of generated videos are shown in App. Fig. 19.

*What are the filtering statistics for different video generation models?* Confirming the trends described above, Fig. 5 reports the pass rates of each model across our four tasks for the GPT-4o filter described in Sec. 3.1. Sora fails all tasks 100% of the time. Kling v1.5 does better, successfully passing pouring 52.6% of the time,

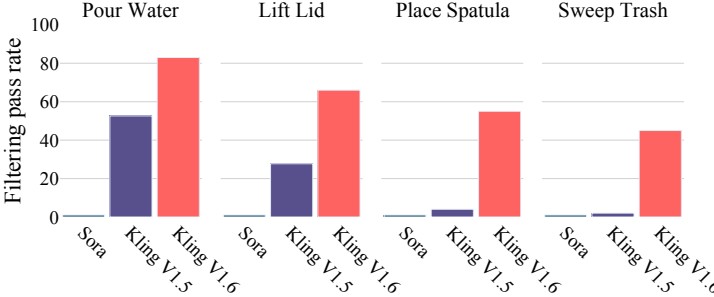

Figure 5: **Filtering statistics.** Kling V1.6 videos have the highest pass rate, demonstrating more accurate adherence to language commands.

lifting 27.7%, placing 4%, and sweeping 2%. Kling V1.6 shows a substantial improvement across tasks, passing pouring 83%, lifting 66%, placing 55%, and sweeping 45% of the time. We noticed that, particularly for the placing and sweeping tasks, even Kling V1.6 frequently generated videos in which the command was not followed. In many cases, the video remained static, and no action was performed.

*How accurate is VLM-based video filtering, and are there any simpler alternatives?* In Tab. 1, we report Pearson correlation coefficients between filtering metrics and human judgments of generation correctness. Our VLM-based filtering achieves strong agreement with human ratings across

| Filtering Metrics | Pour Water | Lift Lid | Place Spatula | Sweep Trash | Average |
|---|---|---|---|---|---|
| **Video-text Consistency** | 0.06 | 0.47 | 0.70 | 0.11 | 0.34 |
| **I2V Subject Consistency** | 0.35 | 0.58 | -0.09 | 0.63 | 0.37 |
| **Querying GPT o1** | **0.91** | **0.91** | **0.91** | **0.66** | **0.84** |

Table 1: **Comparison of video filtering metrics.** Pearson correlation coefficients measure each metric's effectiveness in assessing whether a generated video follows the language command. Querying GPT o1 proves to be most effective.

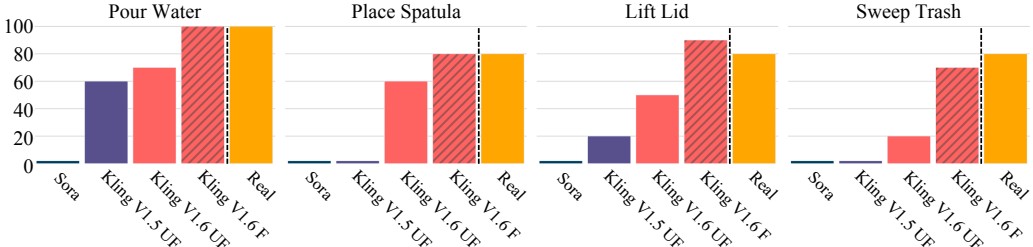

Figure 6: **RIGVid performance vs. video quality.** The dashed lines separate performance on generated videos from real videos. Kling V1.6 produces most reliable videos and leads to highest RIGVid success. Filtered videos perform on par with real ones. UF denotes unfiltered and F denotes filtered.

all tasks, with high correlation values. Most errors made by the VLM-based filter are false negatives—occasionally discarding usable videos, but almost never passing an incorrect one. We also explore the most relevant metrics for our case from a recent benchmark suite for evaluating video generation quality and instruction following, VBench++ (Huang et al., 2024b): (i) video-text consistency measuring the alignment between the command and the generated video (Wang et al., 2023), and (ii) image-to-video (I2V) subject consistency which evaluates whether subjects present in the input image persist throughout the video (Caron et al., 2021). These metrics correlate only weakly or inconsistently with task success and are not reliable for filtering.

*Does higher video quality lead to better robot performance?* To quantify this, Fig. 6 plots RIGVid 's task success across five video sources: unfiltered Sora, unfiltered Kling v1.5, unfiltered Kling v1.6, filtered Kling v1.6, and real human demonstration videos. For each source, we use 10 videos per task. We observe a clear trend: as video quality improves, so does success rate. Sora's unfiltered videos lead to 0% success rate, Kling v1.5 performs better, and Kling v1.6 gives the highest results among all generated videos. Filtering dramatically improves reliability: after discarding failed generations using our automatic approach, success rate with filtered Kling v1.6 videos improves from 80% to 100% on pouring, from 60% to 80% on lifting, from 50% to 90% on placing, and from 20% to 70% on sweeping.

*Can generated videos replace real videos for imitation?* The results in Fig. 6 indicate that, when using filtered Kling v1.6 videos, RIGVid 's performance is similar to that achieved with real human demonstration videos. This finding suggests that, at current model quality, generated videos are already sufficient for visual imitation, substantially reducing the need for manual data collection.

*What causes failure of imitation given high-quality videos?* Aside from one case where the object slipped out of the gripper, all failures on filtered Kling v1.6 videos are attributed to errors in depth estimation. These errors result in inaccurate 6D trajectories and lead to tracking failures. Notably, similar depth estimation issues are also observed in real videos, suggesting that the limitation lies in the depth model itself. App. I provides a detailed analysis of failure cases with qualitative examples.

### 4.3 RIGVID VS. VLM-BASED TRAJECTORY PREDICTION

Video generation is computationally expensive, prompting the question of whether more efficient alternatives can enable robot manipulation without any demonstrations. VLMs offer one potential alternative by predicting simplified task abstractions—goal states (Huang et al., 2023), constraints (Huang et al., 2024a), or reward functions (Patel et al., 2025)—without generating full visual sequences, making them cheaper in computation and inference time. We take the state-of-the-art ReKep (Huang et al., 2024a) method as a representative of this line of work, and compare against it in Fig. 7. In our comparison, RIGVid achieves 85% vs. ReKep's 50% success over four tasks. App. E illustrates ReKep's failures, which we attribute to inaccurate keypoint predictions.

This comparison suggests that, for our tasks and experimental setup, VLM-generated abstractions are compact and may lack the rich, necessary details for successful robot execution. Thus, despite its higher cost, video generation provides crucial supervision rather than being a wasteful expense.

While this result highlights, for our tasks and setup, the additional detail in generated videos supports more reliable execution than the current VLM-based alternative, it does not rule out the possibility that future or alternative VLM-based approaches could close this gap. Our findings suggest that, at present, video generation can provide richer supervision for manipulation compared to this specific VLM-based method, despite its higher computational cost.

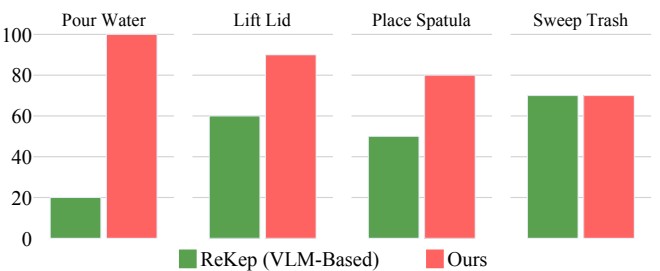

Figure 7: **RIGVid vs. ReKep Success Rates.** RIGVid outperforms SOTA VLM-based trajectory prediction method.

### 4.4 COMPARISON TO ALTERNATIVE TRAJECTORY EXTRACTION METHODS

We investigate the best way to extract trajectory information from videos for the purpose of visual imitation. To this end, we adapted several competitive methods that use different types of tracking to imitate a video without demonstrations. For each method, we describe its inputs and outputs, core approach, our modifications, and the motivation for its inclusion (additional details in App. F).

**Track2Act (Bharadhwaj et al., 2024b) (Tracks-Based)**. This method takes an RGBD image of the initial scene, and a single goal image that specifies the desired final configuration. Since we have no other way to get the goal image, we set it to the last frame of the generated video. Using only this pair of images, Track2Act uses a learned model to predict a dense grid of 2D point tracks, producing pixel-level correspondences between the initial and goal image. These tracks are then lifted to 3D using the depth map from the initial frame and converted into a sequence of 3D object poses via the Perspective-n-Point (PnP) algorithm. We do not finetune their track prediction network, and do not use their residual policy. Track2Act is an attractive alternative as it uses a dedicated track prediction network that operates solely on the start and goal images, without requiring any intermediate frames. However, its main drawback is that the learned track prediction network may not generalize to all scenarios, as evidenced by our experiments and qualitative results.

**AVDC (Ko et al., 2023) (Flow-Based)**. Given an initial RGBD image, task description, and active object mask, AVDC predicts object motion by first generating a task-conditioned video and then computing optical flow between frames. This optical flow is used in an optimization process to reconstruct the object trajectory. In our adaptation, we substitute AVDC's original video generator with our improved model, while preserving all downstream processing. Unlike Track2Act, AVDC leverages optical flow across the entire video, giving it dense temporal correspondences at every pixel and thus many more cues for tracking. It is attractive because it offers a denser input for object tracking. Nevertheless, it is sensitive to cumulative errors in flow estimation, which can degrade the accuracy of the resulting object trajectories.

**4D-DPM (Kerr et al., 2024) (Feature Field-Based)**. This method takes a 3D Gaussian splatting field of the object and a real video of the task, and outputs object trajectories over time. A feature field, similar to NeRF representations, is a continuous mapping from 3D space to high-dimensional feature vectors that capture both geometry and appearance. By aligning the feature field with individual video frames, the method can estimate object trajectory across the video. To build the field, 4D-DPM requires a separate video where the object is captured from all sides. In our adaptation, since 4D-DPM expects a real human demonstration video, we instead use a generated video as the task input video. We modify this method from tracking object part poses to tracking single objects. This approach is compelling because it applies semantic, feature-based reasoning to track objects, capturing entire object structure from video, without relying on explicit correspondences. However, it tends to produce unstable tracking in our experiments, limiting its practicality.

**Gen2Act (Bharadhwaj et al., 2024a) (Generated Goal-Based)**. Gen2Act takes as input an RGBD image of the scene and a task description, and outputs robot actions predicted by a learned policy. In the original formulation, the extracted tracks on the generated video were used to supervise

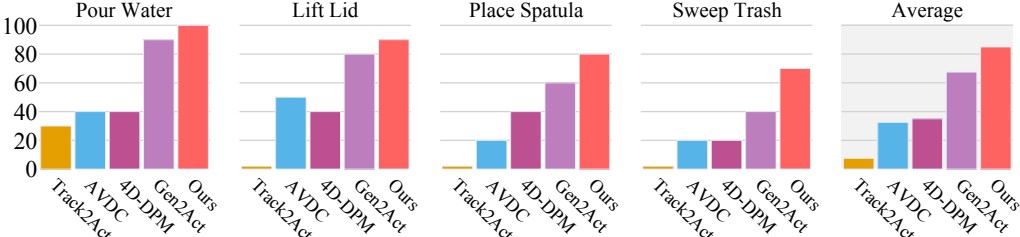

Figure 8: **Comparative evaluation of trajectory extraction methods.** RIGVid consistently achieves higher success across all four tasks; relative improvements are higher as tasks become harder (*i.e.*, from left to right).

behaviour-cloning on a large offline robotics dataset. In our adaptation, we do not use any policy learning. Instead, we extract object tracks from the generated video and directly estimate object poses from these tracks using the initial depth image. This removes any dependence on expert demonstration data or learned policies. Gen2Act is notable for leveraging sparse correspondences extracted from the generated video, enabling task-relevant object motion to be tracked and retargeted without requiring explicit actions. However, when large portions of the object become occluded or undergo significant rotations, many of the tracking points are lost, resulting in too few correspondences to estimate object pose accurately and ultimately causing the tracking to fail.

Fig. 8 shows that RIGVid achieves a success rate of 85.0%, compared to 67.5% for Gen2Act and considerably lower rates for all other baselines. This margin grows with more complex tasks. Methods such as Track2Act (7.5%), AVDC (32.5%), and 4D-DPM (35.0%) rely on point tracks or optical flow, but their performance remains limited, especially as object rotations or occlusions are severe. Gen2Act, which combines video generation with point-based tracking, closes part of the gap but consistently struggles when large portions of the object become untrackable. In contrast, RIGVid's use of 6D object pose trajectory enables robust execution, accounting for the 17.5% improvement over Gen2Act. This advantage persists when more powerful tracking models like Co-Tracker3 (Karaev et al., 2024) are used, as shown in App. G.

Looking at the task-wise breakdown in Fig. 8, RIGVid maintains high success rates even as object depth varies significantly (such as in the lifting task) or when the objects are thin, small, or partially occluded (such as in placing a spatula or sweeping trash). Other methods frequently struggle in these

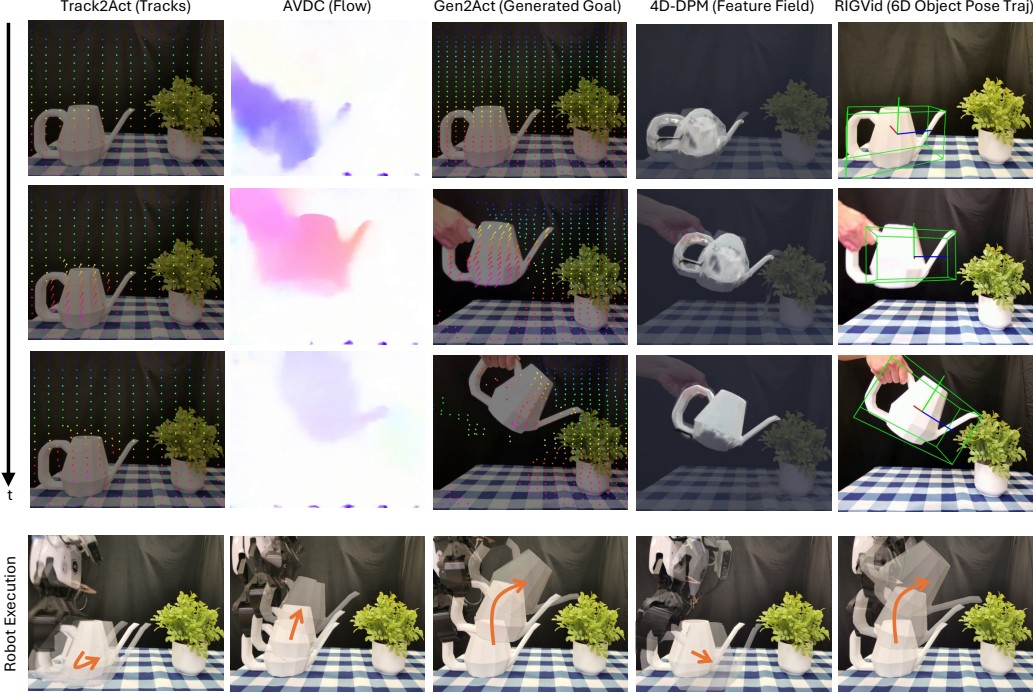

Figure 9: **Analyzing intermediate visual representations.** Only Gen2Act and our 6D Object Pose Trajectory can correctly track the position and rotation of the watering can, leading to a successful execution. Check the description in the main paper for detailed discussions of the failure modes of the alternative methods.

settings, often failing to recover object trajectories when objects become partially hidden or change distance rapidly. The advantage of RIGVid is most pronounced on the most challenging tasks: for both spatula placement and sweeping, RIGVid achieves success rates 20–25% above the next best baseline. These results suggest that the structured 6D pose trajectory not only enables robust tracking under depth changes and occlusion but also scales to scenarios where correspondence methods fail.

Visualizing the outputs in Fig. 9 for the same generated video, we observe the intermediate predictions and resulting robot executions. For Track2Act, the predicted tracks diverge from the true object path, leading to failed execution. Often, the track2act track prediction does not follow the true motion paths, which is the primary source of errors in our experiments. AVDC generates reasonable optical flow in individual frames, but when summed across the entire video, the resulting trajectory is often physically implausible, and the execution fails. We often found that this summing up of object flow across the video leads to small errors that accumulate over the entire video, resulting in faulty object location across the video. Gen2Act yields plausible tracks and leads to successful manipulation. We often found that tracks were accurate, and the resulting trajectory after PnP was also accurate. 4D-DPM exhibits inconsistent tracking performance. While it accurately follows the object in certain segments, the example shown reveals incorrect tracking during the first half of the episode, which ultimately causes the rollout to fail. We often found that the tracking was unstable and very jerky. In contrast, the 6D object pose trajectories produced by RIGVid remain stable throughout the episode and closely match the actual object motion, resulting in successful execution.

### 4.5 TESTING GENERALIZATION

**Embodiment-Agnostic Transfer.** We test RIGVid's generalizability to another embodiment by deploying it on the ALOHA robot for the pouring task (Fig. 10, top left). On this setup, it achieves 80% success, compared to 100% on our default xArm setup.[1] RIGVid also generalizes to a bimanual setup, simultaneously placing a pair of shoes into a box using both arms (Fig. 10, bottom left).

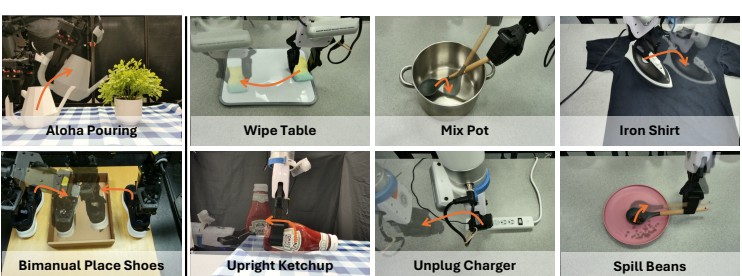

Figure 10: **RIGVid's embodiment-agnostic capabilities and examples on solving complex, open-world tasks.** RIGVid can readily work on ALOHA setup (Zhao et al., 2023) as shown on top left. On the bottom left, RIGVid is retargeted to the bimanual ALOHA setup. On the right, it generates trajectories for diverse manipulation tasks—including wiping, mixing, and ironing—without using any physical demonstrations.

**Extensions to Additional Tasks.** Besides our four main focus tasks, we also obtained promising preliminary results on a larger variety of diverse and challenging manipulation tasks shown in Fig. 10 (right). These tasks include wiping, mixing, and ironing, uprighting a ketchup bottle, unplugging a charger, and rotating a spoon to spill beans. Notably, the latter three tasks involve extreme rotations, which RIGVid can handle successfully.

## 5 CONCLUSIONS

We introduced RIGVid, the first method for robotic manipulation that works from just generated videos. By leveraging recent advances in generative vision and pose estimation, RIGVid enables robots to execute complex tasks entirely from generated video. We extract 6D Object Pose Trajectory from the generated videos and retarget it to the robot, demonstrating a data-efficient approach to robotic skill acquisition. Our analysis shows a correlation between video quality and task success. Additionally, our comparisons with SOTA VLM-based manipulation methods confirm that leveraging dense visual cues from generated videos yields more reliable performance. We also show that RIGVid significantly outperforms competing trajectory extraction methods across a diverse set of tasks, and demonstrate the robustness of our approach to disturbances. Our work represents a step toward enabling robots to learn from the visual knowledge in generative models, reducing reliance on time-consuming data collection.

---

[1]The performance drop stems from inaccurate camera calibration of ALOHA's wrist cameras.

## 6 ACKNOWLEDGEMENT

We thank the members of the RoboPIL lab, and UIUC vision and robotics labs for their valuable discussions and feedback. Unnat would like to especially acknowledge Chen Bao, Homanga Bharadhwaj, Shikhar Bahl, and friends at CMU and Skild for their insightful conversations on learning from videos. We also thank Justin Kerr for his assistance in reproducing the 4D-DPM baseline. This work is partially supported by the Toyota Research Institute (TRI), the Sony Group Corporation, Google, Dalus AI, and an Amazon Research Award, Fall 2024. This article solely reflects the opinions and conclusions of its authors and should not be interpreted as necessarily representing the official policies, either expressed or implied, of the sponsors.

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

# Appendix

We structure the supplement into the following subsections:

We are attaching the source code in the supplementary materials for reproducibility.

## A  BEST PRACTICES FOR VIDEO GENERATION

We found that the following practices lead to reliable video generation: (1) having a clean background without visual distractions, (2) minimizing the number of distractor objects in the scene, (3) ensuring objects are reasonably large and viewed from a natural, human-like perspective, (4) ensuring there is one clearly identifiable task that can be performed, (5) using simple and concise text prompts, and (6) setting the relevance factor to 0.7 with the negative prompt "fast motion" led to the most reliable video generations.

## B  PROMPTING FOR VIDEO FILTERING AND FILTERING STATISTICS

The prompt for GPT o1-based filtering is shown in Figure 16. We provide GPT o1 with the prompt, a video summary—created by vertically concatenating evenly sampled frames from the video—and the language command (e.g., "pour water"). GPT o1 then responds with "Yes" or "No" to indicate whether the task is successfully performed.

## C  MESH-FREE OBJECT TRACKING

We experiment with a mesh-free object tracking version of our method. Specifically, we use BundleSDF (Wen et al., 2023a), which jointly performs 6-DoF object tracking and reconstruction from RGBD observations. For the *pouring* task, we evaluate our method using trajectories obtained via BundleSDF over 10 trials and observe a success rate of (90%), matching our default tracking setup. While the BundleSDF paper reports real-time capabilities, we found that its official implementation takes approximately 30 minutes to process each video in practice, which limits its applicability for real-time deployment. In contrast, our default tracker operates in real-time, enabling closed-loop execution and recovery from disturbances as discussed in Sec. 4.5. While the BundleSDF paper reports real-time capabilities, we observed significantly higher runtimes in practice with the official implementation. We expect that future advances in model-free tracking will address these efficiency bottlenecks, allowing for real-time mesh-free deployment.

## D  SMOOTHING OBJECT TRAJECTORIES

To reduce noise and jitter in the estimated object poses, we apply a moving average filter with a fixed sliding window (centered on each point) to the position and orientation components. Translations are smoothed independently along each axis, while orientation is processed similarly after converting from quaternions to rotation vectors. This approach mitigates abrupt changes, resulting in a more stable and realistic object trajectory with smoother transitions.

## E    ReKep Predictions and Executions

A detailed example of ReKep's keypoint and VLM predictions for pouring task is shown in Fig. 17. The VLM first predicts grasping the watering can at keypoint 1. For the transport phase, it instructs moving keypoint 8 above keypoint 15, while keeping its height above keypoint 7. For the pouring action, keypoint 8 remains above 15 (to place the spout over the plant) and above keypoint 4 (to induce tilting). The resulting robot execution fails. We attribute most ReKep failures to inaccurate keypoint predictions, as shown in Fig. 11. In the lid image, no keypoint appears at the lid handle. In the placing task, keypoints cluster around pan corners. For the sweeping task, the keypoints are generally well-placed, and executions succeeded. Suboptimal initial keypoints lead to inaccurate downstream VLM predictions.

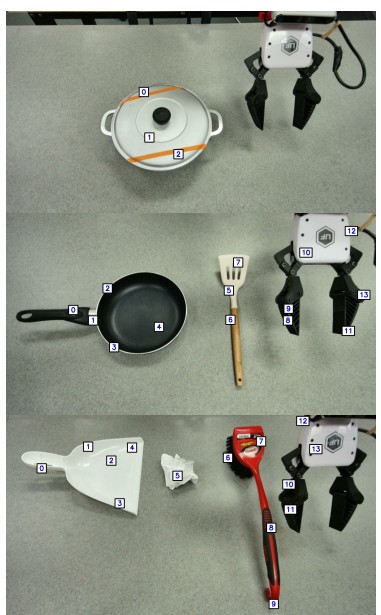

Figure 11: **Examples of ReKep's Keypoint Locations.** The keypoint placements are often suboptimal, except for sweeping task, where the keypoints are reasonable.

## F    Description of Baselines

**Track2Act (Bharadhwaj et al., 2024b):** We adapt Track2Act's procedure to our setup preserving its core idea of object-centric trajectory estimation from point tracks. Track2Act generates a future interaction plan by predicting 2D point trajectories (using a DiT-based diffusion model) between an initial image and a goal image, then recovers a sequence of 3D object transforms via Perspective-and-Point (PnP) (Zhang, 2000).

To integrate this into our pipeline, we use their published checkpoint but modify the input formulation–while the initial image remains identical to our real camera's view, the goal image is taken from the last frame of a generated video rather than being physically captured. We then use PnP on the predicted point tracks along with the initial depth image to estimate the object's rigid motion across frames, thereby defining the end-effector trajectory. We use interpolation between consecutive poses because Track2Act generates only a sparse set of frames, and denser sampling is needed for smooth trajectory estimation and execution. However, we exclude Track2Act's closed-loop residual policy correction, focusing solely on open-loop 6D object-pose estimation and execution. This adaptation allows us to directly evaluate how well a vision-based, open-loop approach generalizes to our setting without additional corrections.

**AVDC (Ko et al., 2023):** The AVDC approach models action trajectories by synthesizing a task-driven video (using a trained text-conditioned video generation model) and using optical flow from GMFlow (Xu et al., 2022) to estimate dense pixel correspondences. It then reconstructs 3D object motion using an optimization step that refines pose estimates based on the tracked flow and depth information. To improve robustness, AVDC also includes a replanning mechanism that re-executes the pipeline when predicted motion stagnates.

Since the trained text-conditioned video generation model did not generalize well to our setup, we use the same generated video as in other experiments to ensure a fair comparison. While we do not employ AVDC's replanning strategy, we predict object poses using a similar optimization framework based on flow and depth information.

**4D-DPM (Kerr et al., 2024):** 4D-DPM is designed to track 3D motion of articulated object parts from a single video. It constructs a 3D Gaussian splatting (Kerbl et al., 2023) representation of the scene to capture object features, then applies GARField (Kim et al., 2024) to cluster the Gaussians into discrete object components. In our adaptation, we modify this to operate on entire objects rather than individual parts. Specifically, we set the clustering parameters to treat the object as a

single entity, ensuring that motion estimation is performed at the object level rather than segmenting it into multiple parts. This allows us to track and execute trajectories for the whole object.

**Gen2Act (Bharadhwaj et al., 2024a):** Gen2Act introduces a video-conditioned policy learning framework that first generates a human video using a video generation model from a scene image and a task description. It then extracts object tracks using BootsTAP (Doersch et al., 2024), and trains a policy using behavior cloning with an auxiliary track prediction loss and offline robot demonstrations. At inference, Gen2Act uses the generated video and the learned policy to predict robot actions.

Our approach presents a simplified adaptation of this framework that removes the need for behavior cloning, and offline demonstrations. Instead of using the extracted tracks as an auxiliary loss, we directly process them for pose estimation. To recover 3D object positions, we leverage an initial depth image corresponding to the scene image, allowing us to obtain depth values for the extracted 2D tracks. We apply RANSAC filtering to remove outlier track points and then use the Perspective-n-Point (PnP) (Zhang, 2000) to estimate the object's 6DoF pose. This adaptation preserves the core idea of leveraging video and track-based signals while eliminating the need for supervised policy learning.

## G LIMITATION OF TRACKING WITH POINT TRACKS

All point tracks fail under extreme rotations, as initially visible points often become occluded. This is a fundamental limitation of any correspondence-based tracking method relying solely on visible surface features. We show this failure in Fig. 12. As the object rotates, most initial points are lost, resulting in insufficient 2D-3D correspondences to solve a stable PnP problem. This degrades pose estimation quality, leading to large drift or abrupt jumps in estimated object motion. Such instability cascades into execution errors, often causing the robot to fail the task altogether. As a result, both variants of Gen2Act—despite stronger tracking backbones like CoTracker—still fail under large out-of-plane rotations. In contrast, RIGVid's model-based 6D tracking handles these situations more robustly, as it uses full-object geometry and SE(3) filtering to maintain stable trajectories.

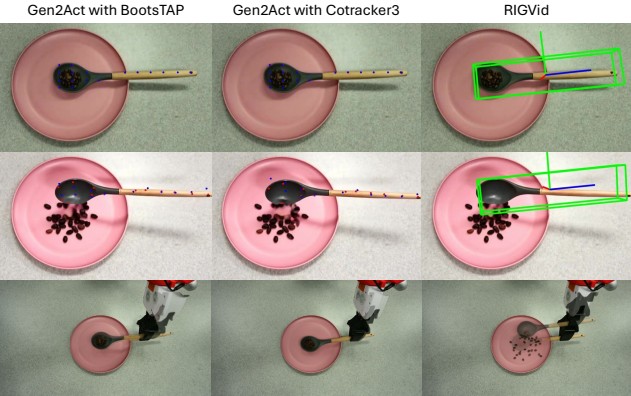

Figure 12: **Gen2Act with BootsTAP, CoTracker, and RIGVid.** Blue points denote the tracked points used for PnP; red points represent the reprojected 3D points. For a good PnP solution, these should align, as seen in the first frame. For Gen2Act, the blue points drift significantly from the red ones in later frames, indicating failure in pose estimation due to tracking loss, which leads to failed robot execution.

## H ADDITIONAL ROBUSTNESS EXAMPLES

Examples of RIGVid's robustness are shown in Fig. 13. In the first row, the robot grasps the object, but due to a misaligned grasp, the object rotates unexpectedly. The robot compensates by rotating it back to the correct orientation and then resumes the planned trajectory, completing the task successfully. In the bottom row, a human perturbs the object during execution while it is held by the robot. RIGVid detects the resulting change in the relative transformation and automatically re-aligns the object before continuing. When the human intervenes a second time, RIGVid again corrects the deviation, resulting in successful task completion.

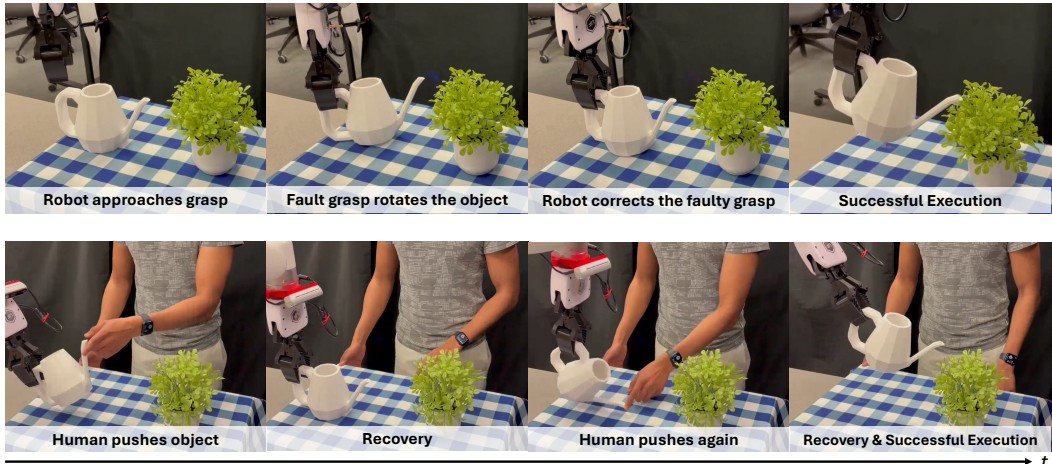

Figure 13: **Additional examples of RIGVid's robustness.** In the top row, RIGVid recovers from a faulty initial grasp by reorienting the object before continuing execution. In the bottom row, it corrects for external disturbances on the object when a human pushes it mid-execution, realigning and successfully completing the task.

## I ERRORS FROM DEPTH ESTIMATION

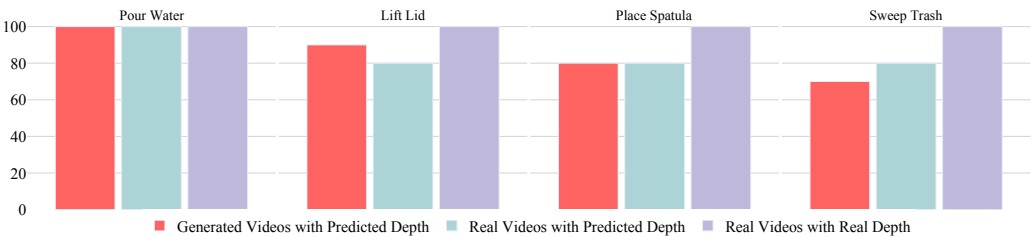

Figure 14: **Impact of Depth Estimation Errors on RIGVid performance.** Errors in monocular depth estimation result in worse performance of generated and real videos. RIGVid achieves perfect success across all tasks with real videos and real depth.

In Fig. 14, we isolate the impact of depth estimation errors. Robot executions on real videos with real depth (captured using an RGBD camera) achieve a 100% success rate, whereas executions from real videos with generated depth result in an 85% average success. Similarly, executions from Kling V1.6-generated videos with generated depth also achieve 85% success, suggesting that the primary source of error lies in monocular depth estimation. Upon inspection, we observe two common undesirable behaviors in the predicted depth: inaccurate depth values and temporal flickering. An example of inaccurate depth is shown in Fig. 15a. In the generated video, when the spatula is brought close to the camera, the depth changes by only 6.8 cm, which is visibly inconsistent with the video and likely much smaller than the real-world change. Inaccuracies also occur in real videos, as shown in the figure—the head of the spatula is estimated to be far from the camera, despite appearing close, revealing another failure mode in monocular depth estimation. Flickering is shown in Fig. 15b. Although the position of the watering can relative to the camera remains nearly unchanged across three consecutive frames, the estimated depth varies significantly. The zoomed-in region on the right shows the can appearing much whiter than on the left, indicating a substantial change in predicted depth. The average depth of the can changes from 40.1 cm to 38.2 cm–a 1.9 cm difference over just 0.066 seconds–which is physically implausible for the generated video. We find similar flickering behavior in real videos as well, where the depth changes from 43.2 cm to 40.9 cm in the given example–a 2.3 cm difference. Since errors in the generated depth are the main source of failure, we also tested removing it entirely by estimating object pose directly from the RGB frames of the

(a) Generated Video

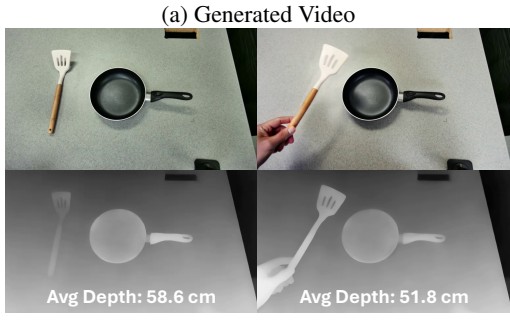

(b) Real Video

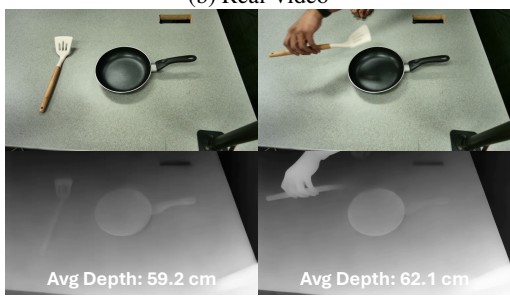

(a) Generated Video

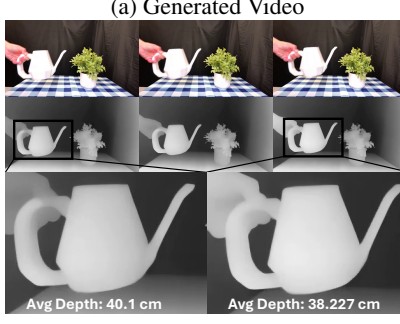

(b) Real Video

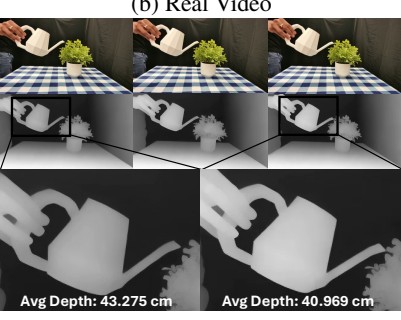

(a) **Errors in Monocular Depth Estimation.** In the generated video (top), the depth of the spatula changes only slightly despite a large visual change. In the real video (bottom), the spatula's head is predicted to lie farther away, contradicting the visual appearance.

(b) **Flickering in Depth Prediction.** We show three consecutive frames of the video and its corresponding predicted depth. The depth of the watering can change noticeably across frames—appearing significantly whiter in the third frame despite minimal actual motion. We observe this behavior in both generated and real videos.

Figure 15: **Combined Figure**: Comparing depth estimation errors (left) and prediction flickering (right) in generated and real videos.

generated video using MegaPose. However, this approach leads to even more unstable and noisy trajectories, as detailed in App. J.

## J CHOICE BETWEEN MEGAPOSE AND FOUNDATIONPOSE

We compare trajectory stability from MegaPose (Labbé et al., 2022) and FoundationPose (Wen et al., 2023b) by computing the translational and rotational RMS jitter. For each method, we apply a Gaussian smoothing filter ($\sigma = 2$ frames) to the raw SE(3) pose sequences, compute the residual between original and smoothed trajectories, and then calculate:

$$\text{jitter}_{\text{trans}} = \sqrt{\frac{1}{N} \sum_{t=1}^{N} \|\Delta \mathbf{t}_t\|^2}, \quad \text{jitter}_{\text{rot}} = \sqrt{\frac{1}{N} \sum_{t=1}^{N} \theta_t^2},$$

where $\Delta \mathbf{t}_t$ is the translational residual at frame $t$, and $\theta_t$ is the angular magnitude (in radians) of the relative rotation $R_{\text{smooth}}^{-1} R_{\text{raw}}$, converted to degrees. Metrics are averaged over ten pouring trajectories from generated videos.

MegaPose yields an average translational RMS jitter of 0.0045m and rotational RMS jitter of 37.47°, whereas FoundationPose achieves 0.0029m translational and 14.31° rotational jitter. This demonstrate that FoundationPose produces significantly smoother and more stable trajectories. Additionally, it allows for real-time tracking during the execution, making RIGVid robust to external disturbances.

## K  COMPARING VIDEO GENERATIVE MODELS

To further assess video quality, we report VBench++ (Huang et al., 2024b) metrics in Table 2 and explain them below. The numbers in the table are scaled $100\times$ for easier interpretation. We collect these metrics on 40 randomly selected and unfiltered videos per model, 10 for each of the four tasks. Kling v1.6 outperformed the other models on most metrics but performed similarly or worse in video-text consistency and dynamic degree. Human evaluations discussed in Sec. 4.2 suggest that the video-text consistency and I2V subject consistency are not reliable indicators of whether a generated video correctly follows a given command. Sora scored high on dynamic degree, likely due to its tendency to drastically alter the scene, resulting in exceptionally large motions. Generated videos from these models and their corresponding metrics are shown in Fig. 18 and further details on these metrics can be found the next section.

**VBench++ Metric Definitions:**

• **Subject Consistency.** Subject consistency describes whether subjects' appearance remain consistent, which is computed by DINOv1 (Caron et al., 2021) similarities across video frames.

• **Background Consistency.** Background temporal consistency by CLIP (Radford et al., 2021) similarities across frames.

• **Motion Smoothness.** Evaluates smoothness of videos by utilizing video frame interpolation model AMT (Li et al., 2023).

• **Dynamic Degree.** Describes whether the video contains large motions as a binary metric.

• **Aesthetic Quality.** Human perceived artistic and beauty value such as photo-realism, layout and color harmony.

• **Imaging Quality.** Assesses the presence of distortion in a video, such as noisiness, blurriness, and over-exposure.

• **Video-Text Consistency.** Text-to-video alignment score calculated by ViCLIP (Wang et al., 2023).

• **I2V Subject Consistency.** Similarity between subjects in input image and each video frame, as well as similarity between consecutive frames. Features are extracted from DINOv1 (Caron et al., 2021).

| Metrics | Video Generation Models | | | Human Demos |
|---|---|---|---|---|
| | Kling V1.6 | Kling V1.5 | Sora | |
| Subject Consistency | **96.34** | 91.66 | 83.09 | 94.91 |
| Background Consistency | **96.64** | 93.97 | 89.34 | 95.00 |
| Motion Smoothness | **99.68** | 99.57 | 99.06 | 99.51 |
| Dynamic Degree | 52.5 | 57.5 | **70.0** | 80.0 |
| Aesthetic Quality | **51.75** | 49.77 | 46.22 | 49.30 |
| Imaging Quality | **72.80** | 71.48 | 68.68 | 72.52 |
| Video-Text Consistency | 22.01 | **22.61** | 21.42 | 21.57 |
| I2V Subject Consistency | **97.88** | 95.96 | 89.09 | 97.89 |

Table 2: **Video generation quality metrics for real human demonstration videos and different models.** Higher values indicate better quality. Kling v1.6 performs comparably to or surpasses other models on most metrics.

Prompt

```
## Instructions
You need to determine whether a specific action occurs in the
given image.
### Image Format
- The image has four rows.
- The action progresses from top to bottom.
### Task
- Examine the image and the provided query.
- Decide if the action described in the query is clearly
depicted.
- A hand should be performing the action. Without it, the
  actions are likely implausible as the action needs to be
  executed by an actor.
- The task has to be done with the specified object in the first
  frame. The video cannot bring in a new object such that there
  are two of the same objects.
### Output Format
Your response must be in the following format
Explain if we see a hand.
Explain if a new object has been brought in that is different
from any objects in the first frame
Explain your reasoning for the given image.
action_done: Yes or No
## Query
Query Task: Pour water from the watering can on the plant
Query Image:
```

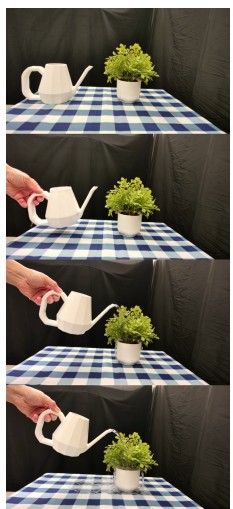

GPT o1 Response

```
We do see a hand holding and tilting the same watering can from the first frame. Water is
visibly flowing onto the plant in the final frame.
action_done: Yes
```

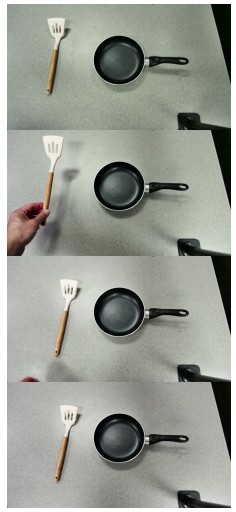

Prompt

```
## Instructions
Same as previous prompt. Instructions omitted for simplicity.
## Query
Query Task: Place the spatula on the pan
Query Image:
```

GPT o1 Response

```
1) Hand present?  Yes, a hand is visible holding the spatula.
2) New object introduced? No, there is no new object introduced; it is still the original
spatula and the original pan.
3) Reasoning:  Although we see a hand holding the spatula, there is no point at which the
spatula is placed on or in the pan. The spatula instead ends up resting on the table next
to the pan.
action_done: No
```

Figure 16: **Examples of prompting GPT o1 to filter generated videos.** We sample frames from the generated video and prompt GPT o1 to assess whether the specified task is performed successfully in the video. The top example passes the filtering, while the bottom does not.

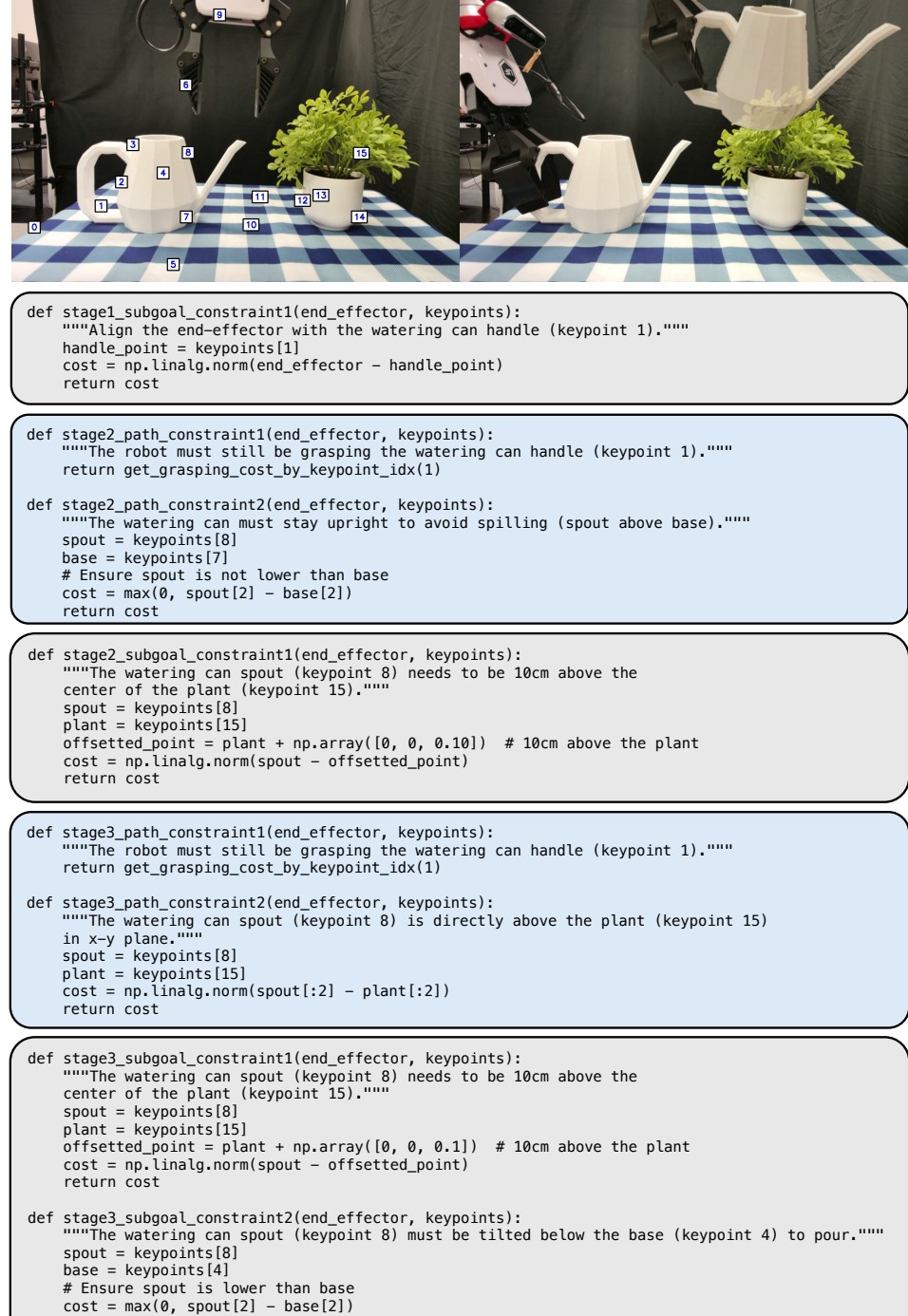

```python
def stage1_subgoal_constraint1(end_effector, keypoints):
    """Align the end-effector with the watering can handle (keypoint 1)."""
    handle_point = keypoints[1]
    cost = np.linalg.norm(end_effector - handle_point)
    return cost
```

```python
def stage2_path_constraint1(end_effector, keypoints):
    """The robot must still be grasping the watering can handle (keypoint 1)."""
    return get_grasping_cost_by_keypoint_idx(1)

def stage2_path_constraint2(end_effector, keypoints):
    """The watering can must stay upright to avoid spilling (spout above base)."""
    spout = keypoints[8]
    base = keypoints[7]
    # Ensure spout is not lower than base
    cost = max(0, spout[2] - base[2])
    return cost
```

```python
def stage2_subgoal_constraint1(end_effector, keypoints):
    """The watering can spout (keypoint 8) needs to be 10cm above the
    center of the plant (keypoint 15)."""
    spout = keypoints[8]
    plant = keypoints[15]
    offsetted_point = plant + np.array([0, 0, 0.10])  # 10cm above the plant
    cost = np.linalg.norm(spout - offsetted_point)
    return cost
```

```python
def stage3_path_constraint1(end_effector, keypoints):
    """The robot must still be grasping the watering can handle (keypoint 1)."""
    return get_grasping_cost_by_keypoint_idx(1)

def stage3_path_constraint2(end_effector, keypoints):
    """The watering can spout (keypoint 8) is directly above the plant (keypoint 15)
    in x-y plane."""
    spout = keypoints[8]
    plant = keypoints[15]
    cost = np.linalg.norm(spout[:2] - plant[:2])
    return cost
```

```python
def stage3_subgoal_constraint1(end_effector, keypoints):
    """The watering can spout (keypoint 8) needs to be 10cm above the
    center of the plant (keypoint 15)."""
    spout = keypoints[8]
    plant = keypoints[15]
    offsetted_point = plant + np.array([0, 0, 0.1])  # 10cm above the plant
    cost = np.linalg.norm(spout - offsetted_point)
    return cost

def stage3_subgoal_constraint2(end_effector, keypoints):
    """The watering can spout (keypoint 8) must be tilted below the base (keypoint 4) to pour."""
    spout = keypoints[8]
    base = keypoints[4]
    # Ensure spout is lower than base
    cost = max(0, spout[2] - base[2])
    return cost
```

Figure 17: **ReKep's output for the pouring task and the resulting robot execution (top-right).** The VLM predicts to grasp at keypoint 1, move keypoint 8 above 15 and 7 during transport, and above 15 and 4 for pouring—leading to failed execution.

### SORA

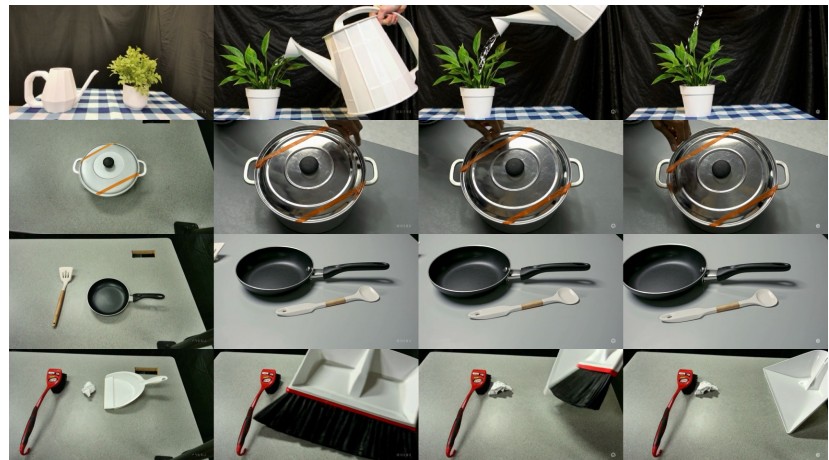

VT Const : 0.267
I2V Subj. Const : 0.887
Subj. Const : 0.808

VT Const : 0.221
I2V Subj. Const : 0.792
Subj. Const : 0.746

VT Const : 0.208
I2V Subj. Const : 0.930
Subj. Const : 0.915

VT Const : 0.218
I2V Subj. Const : 0.977
Subj. Const : 0.839

### Kling AI v1.5

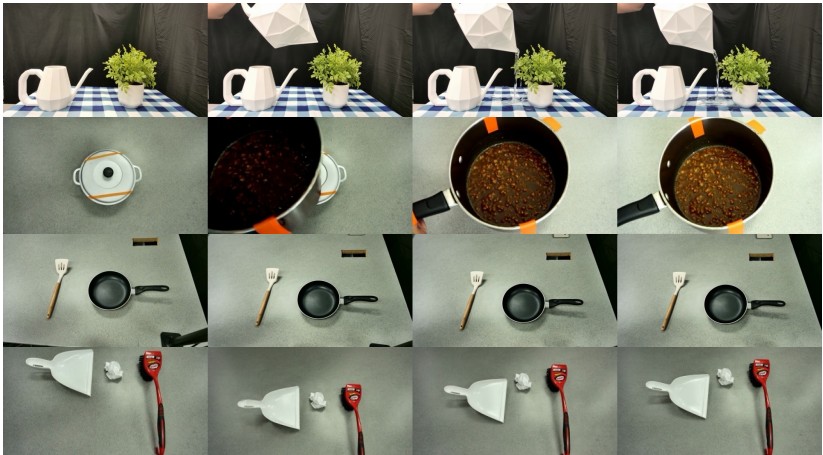

VT Const : 0.244
I2V Subj. Const : 0.989
Subj. Const : 0.936

VT Const : 0.195
I2V Subj. Const : 0.978
Subj. Const : 0.731

VT Const : 0.231
I2V Subj. Const : 0.989
Subj. Const : 0.982

VT Const : 0.201
I2V Subj. Const : 0.865
Subj. Const : 0.965

### Kling AI v1.6

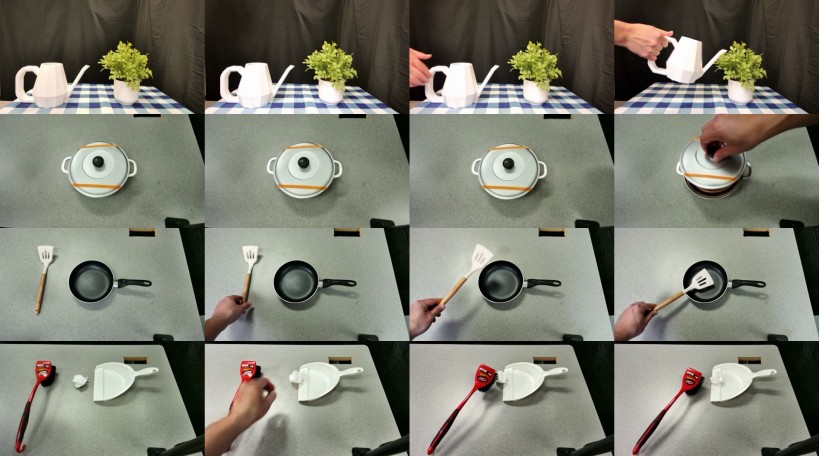

VT Const : 0.217
I2V Subj. Const : 0.995
Subj. Const : 0.975

VT Const : 0.208
I2V Subj. Const : 0.964
Subj. Const : 0.969

VT Const : 0.245
I2V Subj. Const: 0.9965
Subj. Const: 0.965

VT Const : 0.188
I2V Subj. Const : 0.955
Subj. Const : 0.951

Figure 18: **Qualitative Comparison of Different Video Generative Models.** Videos from the three video generation models are shown using evenly sampled frames, along with VBench++ (Huang et al., 2024b) metrics: video-text consistency, image-to-video subject consistency, and subject consistency. Kling v1.6 scores highest on these metrics, followed by Kling v1.5 and then Sora.

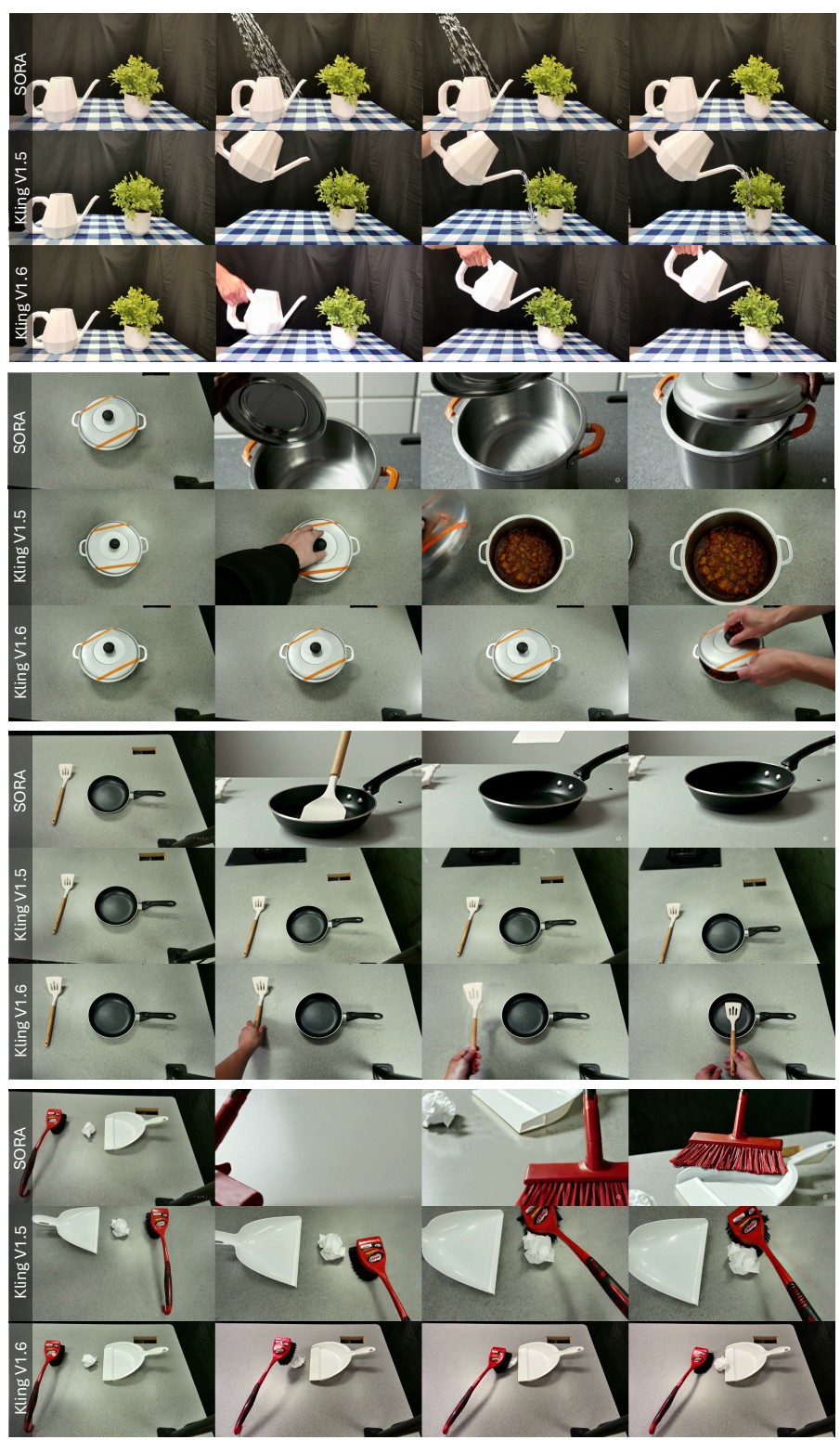

Figure 19: **Qualitative comparison of video generation.** Sora-generated videos often alter the scene layout and objects. Kling V1.5 produces more plausible results but includes physically implausible elements. Kling V1.6 better preserves scene fidelity and closely follows the human command.

