# OpenReview forum: "Robotic Manipulation by Imitating Generated Videos Without Physical Demonstrations"
_ICLR.cc/2026/Conference — ICLR 2026 Poster_

### Official Review · Reviewer_Jtgy · 2025-10-24

**Soundness:** 4
**Presentation:** 4
**Contribution:** 4
**Rating:** 6
**Confidence:** 5

**Summary:**

Summary:

RIGVid enables a robot to execute real‑world manipulation by imitating AI‑generated videos—without physical demonstrations or robot‑specific training. Given a language command and a scene image, the system (1) generates a candidate video, (2) filters low‑quality generations with a vision‑language model (VLM), (3) extracts a 6‑DoF (6D) object‑pose trajectory, and (4) retargets that trajectory to different robot embodiments for closed‑loop execution.

Contributions:

End‑to‑end from generated video: First framework to execute real‑world manipulation directly from generated videos, requiring no physical demos or robot‑specific training.

Generated videos as supervision: Empirically shows that high‑quality, VLM‑filtered generated videos can substitute for real demonstrations in visual imitation.

Simple, effective recipe: Video generation → VLM filtering → 6‑DoF pose extraction → embodiment‑agnostic retargeting; this pipeline outperforms SOTA alternatives based on VLM abstractions, point/flow tracking, feature fields, or generated goal images.

Closed‑loop, robot‑agnostic control: Robust to disturbances via real‑time re‑tracking and recovery.

Insights:

Filtering is effective. VLM‑based filtering correlates strongly with human judgments and, when it errs, tends to be conservative (false negatives), which is the safer failure mode.

State extraction is key. The 6‑DoF pose trajectory extracted from generated videos serves as a compact, transferable dataset format and appears to be the minimal supervision needed for successful grasping.

Overall assessment:

This is a strong paper with a few remaining questions. If the authors are willing to address them, I would be happy to raise my score accordingly.

**Strengths:**

Closed-loop robustness. Real-time re-tracking and an explicit backtracking rule (≥3 cm or ≥20° deviation) enable recovery from perturbations—practically valuable and often missing in imitation-from-video work.

Thorough comparative evaluation for trajectory extraction. Benchmarks span tracks (Track2Act), flow (AVDC), feature fields (4D-DPM), and generated-goal supervision (Gen2Act). The 6D-pose rollout consistently performs best, especially on harder cases (thin/occluded objects, depth discontinuities).

Useful ablations on video quality and filtering. Better generators (e.g., Kling v1.6) and automatic VLM-based filtering materially improve success; filtered generations approach real-video performance—an actionable result for the community.

 Demonstrates portability (xArm7 → ALOHA, including a bimanual vignette) via an object-to-end-effector transform, suggesting the representation is robot-agnostic.

**Weaknesses:**

Scale and statistical power. Main quantitative results cover only four tasks and, for some comparisons, use 10 videos per task per source with human-judged success. That’s limited for strong claims like “on par with real demonstrations,” and no confidence intervals are reported.

Compute and latency under-specified. The pipeline generates and filters videos (up to five attempts), runs depth estimation and 6D tracking, and then executes in closed loop. The paper acknowledges high cost but does not quantify wall-clock time, attempts-per-success, or throughput—metrics that matter for practical deployment.

**Questions:**

Question 1 :What do generated videos add to robotic manipulation?

My current understanding is that generated videos provide a tighter visual grounding between the object and the robot than trajectory-only supervision. If the robot is ultimately learning a pick-and-place trajectory, why not supervise directly on trajectories? My hypothesis is that VLM-based filtering removes irrelevant content and that generated videos interface more naturally with VLMs than trajectories do.

Request: Please show an experiment that visualizes which generations are filtered out and why. Even better, compare (a) VLM filtering on generated videos vs. (b) an LLM (text-only) filtering of candidate trajectories, holding downstream training constant. If (a) > (b), that would substantiate the claim—and I’m inclined to raise my score if I can see this result.


Question 2:
Please include more qualitative failure cases of the full method (e.g., failure modes in re-tracking, depth errors on transparent/reflective objects, backtracking oscillations, or embodiment-transfer edge cases), along with brief diagnoses and suggested fixes.

---

> ### Author Response · Authors · 2025-11-21
> **Rebuttal by Authors**
>
> **Scale and statistical power…**\
> Real-world robotic evaluation is resource-intensive. Running just the ablations in Sec 4.5 took us around 30 hours, where a person had to be present to manually conduct the study. Our evaluation, with 10 trials per task across multiple baselines and ablations, is as extensive and standard for real-robot manipulation [1,2].  Within our constraints, we present the following for a thorough understanding of our method:
> * Quantitative results on 4 diverse tasks against 5 SOTA baselines.
> * Qualitative generalization to additional tasks and a different robot.
> * Error breakdown to provide insights into the current bottlenecks and future investigations.
> * Key insights such as filtered gen videos can be as effective as real videos, and the video-based method outperforms the VLM-based method.
>
> We believe that these will be valuable for researchers who are trying to leverage gen videos for robotics. To further quantify uncertainty as requested by the reviewer, the 95% confidence intervals for task success are 72–100 for pouring, 60–98 for lifting, 49–94 for placing, and 40–89 for sweeping.
>
> **Compute and latency under-specified…**\
> Video gen takes approximately 1 min. VLM-based filtering requires less than 2 sec. Depth estimation takes approximately 1 min, and 6D tracking requires around 10 sec. Our automatic generation typically produces a plausible video within 1-2 attempts (as reflected in our filtering success rates in Table 1). Therefore, end-to-end trajectory extraction generally completes within 2-5 min. We will add this discussion in the App.
>
> **Examples of filtered out generations…**\
> * Unnecessary introduction of objects [link1](https://drive.google.com/file/d/1HQhi67fZYS8DM3jchMzdRDnJOpwXi1At/view?usp=sharing)
> * Not executing the task [link2](https://drive.google.com/file/d/1cPkdVn4kxOn25oHu9dOwLZEh56iR4DGo/view?usp=sharing)
> * Change of object [link3](https://drive.google.com/file/d/11ib0q5ZGVTOfcM_89uGvY8ogYVTehnNd/view?usp=sharing)
> * Attempting, but not achieving the task [link4](https://drive.google.com/file/d/10jKOY5Pb6PetMoQcv1dq0JHmNzcksW-7/view?usp=sharing)
>
> **Video generation advantages to robotic manipulation…**\
> To address the reviewer's concern, we conduct an exp to compare video-based VLM filtering against a text-only LLM filtering. To instantiate such a text-only filtering. We provide the LLM with a trajectory description derived from the same gen video, using the following text-only inputs to GPT-5 model:
> 1. *Absolute pose sequence*: A discretized list of 6D object poses over time, serialized into text and accompanied by the command.
> 2. *Relative pose sequence*: The same poses expressed relative to the initial pose, highlighting motion rather than absolute pose.
> 3. *Dense caption*: A detailed textual description of the video produced by an off‑the‑shelf captioner from Yang et al. [3].
>
> The LLM is asked whether, given this non-visual description, the trajectory satisfies the command. Across pouring and lifting, this text-only baseline achieves Pearson correlations of 0.01/0.02 (absolute poses), 0.02/0.04 (relative poses), and 0.12/0.05 (dense captions), all substantially lower than the video+VLM correlation of approximately 0.91. This gap shows that, even with carefully engineered textual trajectory descriptions, a text-only LLM assesses success far less reliably than a VLM operating directly on video frames.​
>
> **Qualitative failure cases…**
> * *Dynamics difference between gen video and real-world*: For the cleaner, there are cases where the task is completed in the gen video but not fully executed by the robot, due to mismatched object dynamics between video and reality. Fine-tuning the trajectory in sim could help resolve this issue [link5](https://drive.google.com/file/d/1F9uQBiFq7ioBJOjrFCH-xkoHZcfjoTjz/view?usp=sharing)
> * *Imperfect depth on shiny/reflective surfaces*: We observe failure cases from tracking caused by imperfect depth estimation on shiny or reflective surfaces. Using recent learning-based stereo methods (e.g., Wen et al. [4]) can yield more reliable depths in such cases [link6](https://drive.google.com/file/d/124H46edIAU6EoqTicDU5kWxn8iugC6nz/view?usp=sharing)
> * *Monocular depth estimation errors*: Some errors arise directly from inaccuracies in video depth prediction, and advances in these models are likely to mitigate this issue [5] [link7](https://drive.google.com/file/d/1IbLE0-pVeZBQxFoU6OQLShI9yGSpuZEl/view?usp=sharing)
> * *Shaky trajectories*: Although not a failure mode, we occasionally observe jittery object trajectories and robot motion. This issue can be mitigated by applying more advanced trajectory smoothing, like from a Kalman filter [link8](https://drive.google.com/file/d/1GvNdPnF-GpeYD5GKmXIw_kHbyr4Eb7xo/view?usp=sharing)
>
> [1] Huang et al. VoxPoser, CORL 2023\
> [2] Nair et al. R3M, CORL 2022\
> [3] Yang et al. Vid2Seq, CVPR 2023\
> [4] Wen et al. FoundationStereo, CVPR 2025\
> [5] Lin et al. Depth Anything 3, 2025

---

> > ### Comment · Reviewer_Jtgy · 2025-11-22
> >
> > Thanks for your detailed feedback.
> >
> > I think most of my concern has been addressed.
> >
> > Just a quick question, I don't understand why Pearson correlations  is used as the metric for filtering.
> >
> > Can you provide more background about this metric and some related paper so that i can learn more about this？

---

> > > ### Author Response · Authors · 2025-11-23
> > >
> > > We use Pearson correlation to evaluate the effectiveness of a filtering mechanism, not as the filtering criterion itself. The filtering mechanisms (such as the VLM + video frames in our work and the LLM-based approach recommended in the review) make binary decisions (Yes/No) about whether a generated video satisfies the criteria. Pearson correlation then measures the alignment between these mechanism decisions and human judgments (i.e., the ground truth).
> > >
> > > As requested, related papers: Huang et al. [1] use Pearson correlation to validate automatic video evaluation metrics against human judgment. In NLP, too, metrics for text quality (such as BLEURT [2]) are evaluated using Pearson correlation with human judgments to ensure that automatic evaluations match human preferences.
> > >
> > > We appreciate the reviewer's attention to detail and active engagement. We hope this response helps everyone understand the intuition behind using Pearson correlation as the metric for the effectiveness of filtering mechanisms.
> > >
> > > [1] Huang et al. VBench, CVPR 2024\
> > > [2] Sellam et al. BLEURT, ACL 2020

---

> > > > ### Comment · Reviewer_Jtgy · 2025-11-23
> > > >
> > > > Thanks!

---

### Official Review · Reviewer_bPbv · 2025-10-31

**Soundness:** 2
**Presentation:** 2
**Contribution:** 2
**Rating:** 4
**Confidence:** 3

**Summary:**

This paper proposes RIGVid, a system that enables robots to imitate AI-generated videos instead of real demonstrations. Using video generation, VLM-based filtering, and 6D pose tracking, the robot executes manipulation tasks directly from synthetic examples. Experiments show synthetic videos can match real demonstrations and outperform keypoint-based and optical-flow baselines. The work suggests high-quality generated video can serve as effective supervision for robotic manipulation, reducing the need for real data.

**Strengths:**

1. The method requires no robot-specific training or real demonstrations, showing strong potential for scalable imitation learning directly
from generated videos.
2. The experimental analysis is thorough and insightful, revealing key factors such as the critical role of depth estimation in successful video-based control.

**Weaknesses:**

1. The paper suffers from dense formatting and uneven space allocation: related work and baseline descriptions are disproportionately long, while the core experimental section is compressed and difficult to follow. This tight layout and aggressive space packing visibly violate ICLR formatting expectations and reduce readability. A clearer narrative structure and more balanced discussion would improve clarity and presentation quality.

2. Although the method follows a different paradigm than Vision-Language-Action (VLA) models, a direct discussion or empirical comparison is still needed. This would contextualize the contribution and help readers understand how generated-video-based imitation currently stands relative to dominant VLA pipelines, clarifying strengths, limitations, and future directions for this alternative trajectory.

**Questions:**

How well does this method generalize? If applied to a scene that has not been tuned or calibrated, is it still likely to succeed?

---

> ### Author Response · Authors · 2025-11-21
> **Rebuttal by Authors**
>
> **Paper suffers from dense formatting…**\
> We thank the reviewer for the suggestion. We will improve the writing and rebalance the space allocation as recommended. In particular, we will relocate some ablations to the appendix, which will enable us to make the main paper less dense.
>
> **Comparison to VLAs…**\
> On our ALOHA pouring task, RIGVid achieves 100% success, whereas the VLA policy pi0, run zero-shot, fails to make progress and instead produces unstable shaking, resulting in 0% success. This highlights that current generalist VLAs struggle when directly deployed on new setups without fine-tuning.\
> Conceptually, VLAs such as OpenVLA directly map images and language to actions and are trained on massive robot-specific datasets (e.g., about 970k episodes and roughly 20k GPU hours of pretraining for OpenVLA followed by fine-tuning), whereas RIGVid rather composes off-the-shelf video generation, depth estimation, pose tracking, and grasping modules at inference time, enabling zero-shot deployment. RIGVid is explicitly designed to avoid three bottlenecks: (1) No large-scale end-to-end VLA pretraining; (2) No task-specific robot demonstrations or finetuning, since it relies solely on generated videos; (3) No embodiment-specific adaptation, because it works with 6D object pose trajectories that can be retargeted across embodiments (e.g., xArm and (bi)manual ALOHA). As a tradeoff, RIGVid is less suitable for tasks that lack a trackable trajectory, such as manipulating highly deformable objects.
>
> **Does RIGVid generalize…**\
> Our method is zero-shot and does not require any finetuning, enabling generalization across different scenarios, embodiments, and from single to bimanual tasks. We demonstrate this in Sec. 4.5 through successful transfer to the ALOHA robot (80% success on pouring) and bimanual setup (placing shoes), as well as extensions to diverse tasks, including wiping, mixing, and ironing. However, depth calibration between the predicted and real depth maps is an integral component of the pipeline. This calibration requires only the initial depth observation, which is present in most robotics setups.

---

> > ### Author Response · Authors · 2025-11-23
> >
> > We are pleased to inform that we have addressed the formatting concerns raised, as can be seen in the updated PDF. Following the discussion period guidelines, which allow 10 pages (versus the 9-page submission), we have revised the paper to eliminate the dense formatting and improve readability.
> >
> > We appreciate the reviewer's suggestions in helping us improve the manuscript.

---

### Official Review · Reviewer_uft8 · 2025-11-01

**Soundness:** 3
**Presentation:** 3
**Contribution:** 3
**Rating:** 6
**Confidence:** 4

**Summary:**

This paper pilotly explores using purely AI-generated videos for robots to perform tasks. The pipeline is very simple but effective: 1. Generate videos according to task instructions; 2. Lift the videos into 3/4D; 3. Performation robot pose estimation on the generated point cloud sequences. Extensive experiments are conducted.

**Strengths:**

1. The idea is insightful. In fact, the reviewer personally also believes in the proposed solution path.
2. The experiments are superior, with extensive ablation studies and real-world evaluations.

**Weaknesses:**

1. Ablations on 3D reconstruction are missing. How about using other SOTA 4D reconstruction methods, instead of just monocular depth estimation? Will it improve the performance?
2. Similarly, the tracking method also needs to be ablated. Especially, what if we just use ICP on the generated sequences? This may help to better evaluate the quality of the generation.
3. If some robot data is used to finetune the video-gen (instead of directly testing), will it improve the performance?
4. Open question: If the quality of the previous video-gen is low, are there any possible designs to post-refine the result, maybe from another action policy?

**Questions:**

Although the proposed method itself is not very impressive, the problem that the paper aims to explore is meaningful. The experiments are very comprehensive, making this paper a good *study* research, instead of a *method* paper.

Therefore, as for the current submission, not only the title but also the way to present the paper should be adjusted to a "study"-style paper, as there is no very novel framework is proposed, and no finetuning is provided, but only different video-gen models are tested.

Moreover, some more experiments mentioned in the weaknesses are expected to be provided to strengthen the paper.

---

> ### Author Response · Authors · 2025-11-21
> **Rebuttal by Authors**
>
> **3D reconstruction ablations and 4D reconstruction methods…**\
> Heavier 4D reconstruction currently does not offer a clear cost–benefit advantage in our setting. It is substantially slower than our depth-based pipeline and, in our experiments, does not improve task success. In early experiments, we replaced monocular depth estimation with Monst3r [1], a recent 4D reconstruction method, and found that while the reconstructed geometry was generally reasonable, the end-to-end pipeline was more than 6× slower than our depth-based approach, making it impractical for our setting. Moreover, on our tasks, we did not observe any improvement in manipulation success over the monocular-depth version, so we chose the latter as the default for the main experiments. Finally, our 4D-DPM baseline already instantiates a 4D feature-field reconstruction and tracking pipeline, and its lower success rates compared to RIGVid further suggest that, in our regime, heavier 4D reconstruction does not yet translate into better manipulation performance. For example, on the pouring task, 4D-DPM achieves a 40% success rate compared to RIGVid's 100%, with similar trends observed across other tasks. However, combining faster 4D methods with our framework remains a promising direction for future work
>
> **Tracking methods ablations and using ICP…**\
> We conduct extensive tracking ablations in Section 4.4, comparing four different tracking paradigms against four baselines: Track2Act (track-based), AVDC (flow-based), 4D-DPM (feature-field-based), and Gen2Act (generated goal-based). RIGVid achieves 85.0% success compared to 67.5% for the next-best baseline (Gen2Act), with advantages most pronounced on challenging tasks involving thin objects or extreme rotations.
> As recommended by the reviewer, we additionally experimented with point-to-plane colored ICP on 10 filtered videos. ICP loses tracking in 6 out of 10 trials. On successful trials, its tracking exhibits significantly higher jitter: translational RMS jitter of 0.012m and rotational RMS jitter of 40.7°, compared to FoundationPose's 0.0029m and 14.31° (jitter computed as described in App. J). This instability aligns with prior community observations [2], which note that ICP "suffers from low accuracy and slow runtime, limiting its applicability."​
>
> **Robot data to finetune the video-gen…**\
> Using robot data to finetune video generation models would likely improve downstream task performance, as demonstrated in prior works [3, 4]. However, this requires substantial robot data collection: Du et. al. [3] use 200,000 videos, and Liang et. al. [4] require 727 videos per single task. Additionally, storing and training on such large video datasets incurs substantial computational and storage costs. Our core contribution is eliminating the need for any robot demonstrations, whether for video generation, fine-tuning, or policy training. This zero-shot capability enables execution on novel tasks without data collection overhead. While incorporating robot data is a promising direction, it fundamentally shifts the problem setting from our focus on demonstration-free manipulation.
>
> **Post refining results…**\
> We implement several measures to post-refine our results and to ensure that the trajectories from generated videos are executable. First, our VLM-based filtering (Sec. 3.1) automatically rejects videos that do not follow commands, achieving strong correlation with human judgments (Table 1). Second, we apply trajectory smoothing to remove abrupt pose changes (App. D).
> For task-level refinement, policies trained on limited robot data could correct minor trajectory errors, similar to Bhardwaj et al. [5]. However, this again requires robot demonstrations, which conflicts with our zero-shot objective.
>
> **Study-style paper…**\
> We respectfully disagree that our method is not novel. RIGVid is the first framework that allows robots to execute complex manipulation tasks solely from generated videos. The core contribution is this novel framework and our specific methodological solutions, like the depth-based scale correction and VLM-based filtering (essential to make RIGVid work with generated videos) are of value to the community that is investigating the use of generated video for robotics.
> We will, however, clarify this dual nature in the paper: we will more prominently frame RIGVid as a novel framework with specific methodological contributions, while also highlighting the empirical comparisons across video generators as a systematic study within this new paradigm, in line with the reviewer’s suggestions.
>
> [1] Zhang et al. MonST3R, ICLR 2025\
> [2] Liu et al. Deep Learning-Based Object Pose Estimation: A Comprehensive Survey, IJCV 2025\
> [3] Du et al. Learning Universal Policies via Text-Guided Video Generation, NeurIPS 2023\
> [4] Liang et al. Dreamitate, CORL 2024\
> [5] Bhardwaj et al. Track2Act, ECCV 2024

---

### Official Review · Reviewer_Tugp · 2025-11-02

**Soundness:** 3
**Presentation:** 3
**Contribution:** 2
**Rating:** 4
**Confidence:** 4

**Summary:**

This paper presents RIGVid, a demonstration-free robotic manipulation framework that enables a robot to execute tasks purely by imitating AI-generated videos. Given a textual command and a single RGB-D observation, the system generates a video via a diffusion model, filters it with a VLM (GPT-4o) for semantic correctness, estimates per-frame depth, extracts a 6D object pose trajectory using FoundationPose, and retargets this trajectory to the robot for execution. The method is evaluated on four real-world manipulation tasks (pouring, lifting, placing, sweeping) and compared against several strong baselines including ReKep, Track2Act, Gen2Act, and optical-flow or feature-field-based trackers. Experiments demonstrate that filtered generated videos can yield comparable success rates to real human videos, supporting the claim that synthetic visual demonstrations can replace physical ones under certain conditions.

**Strengths:**

The paper proposes a demonstration-free robotic manipulation framework that relies solely on generated videos. While the approach is currently constrained by the capability of existing video generation models, it points toward a compelling direction: as world-model generators such as SORA and Veo continue to evolve, this paradigm could become a practical and scalable way to get rid of robot teleoperation or imitation data.

The system is solidly engineered. It combines existing modules for depth estimation and 6D pose tracking in a sensible way to get a high-level “pose flow” that drives the robot’s actions. While I don’t think pose is the final or ideal representation for manipulation (as I’ll mention later), this design makes sense at the current era of the  co-evolving embodied AI and generative world models. At least for the four manipulation tasks shown in the paper, the whole pipeline works coherently and delivers consistent results.

The experiments cover a diverse set of real-world manipulation tasks and include detailed ablations on video quality, filtering accuracy, and trajectory extraction methods. The results provide convincing quantitative and qualitative evidence for the framework’s effectiveness.

The paper is clearly written and easy to follow. The figures and descriptions effectively illustrate the overall pipeline

**Weaknesses:**

The use of FoundationPose requires access to the CAD or mesh models of target objects, or at least works significantly better when such models are provided. This requirement limits the applicability of the system in unstructured or open-world environments where object models are unavailable. In addition, while modern video generators such as Kling can potentially produce videos containing deformable objects, FoundationPose cannot reliably track non-rigid shapes, which further constrains the border usage of the proposed pipeline.

Lack of details on handling pre- and post-grasp phases. Since the method uses the object trajectory directly as the source for retargeting, the robot gripper is assumed to remain fixed at the grasp position. It is unclear how the system handles the pre-grasp phase (approaching and aligning for grasping) and the post-manipulation phase (safely releasing or retracting the gripper). These should ideally be integrated into the overall policy for realistic and safe execution.


The paper does not evaluate any explicit recovery behavior after execution failures. It would be helpful to know how long the recovery takes, and whether the system (or the video generator such as Kling) can re-generate corrective videos to handle failed trials or unexpected conditions.


Current video generation models are limited by the duration of the generated content. The paper does not analyze how such limitations affect tasks with longer execution horizons, where the manipulation may take more time than the video generator can represent.

**Questions:**

The current pipeline extracts trajectories based on human-hand motion in the generated videos.
Have the authors considered directly generating robot gripper poses from Kling or similar models, which might align better with robotic control?

---

> ### Author Response · Authors · 2025-11-21
> **Rebuttal by Authors**
>
> **Strict requirement of CAD model…**\
> We do not strictly require object meshes, as discussed in Sec. 3.2 and App. C. Mesh-free methods like BundleSDF [1] can track objects without requiring a mesh; however, they remain slow for real-time tracking. We demonstrate that mesh-free tracking using BundleSDF achieves a 90% success rate on the pouring task. With advances in mesh-free tracking, these efficiency bottlenecks will be addressed. Alternatively, we could also use image-to-3D generation models [2] to create object meshes, then apply tracking, eliminating the need for a pre-computed mesh.
>
> **Cannot track non-rigid shapes…**\
> RIGVid can be extended to handle such objects by tracking keypoints instead of poses, an approach that has already shown success in LLM-based trajectory execution [3]. We agree that this is a valuable direction for future work and will include a discussion on this in the updated version.
>
> **Pre-grasp and post-execution…**\
> We use AnyGrasp to obtain the grasp pose, then command the robot gripper to first move to a pre-grasp pose (same orientation, offset distance from the object). The gripper then approaches the grasp pose and closes. For post-manipulation, we currently maintain the closed gripper state throughout task execution. However, it can be extended to open grippers when specific constraints are met, as demonstrated by Huang et. al. [3].
>
> **Evaluation of recovery behaviors…**\
> Our robustness implementation enables instantaneous recovery from execution deviations. Recovery time depends on deviation magnitude. For instance, objects pushed far from the trajectory require longer repositioning. For task-level failures, we can integrate a VLM-based verification step post-execution. If the task is incomplete, the system can regenerate videos and retry, similar to approaches in VLM-based manipulation [4].
>
> **Length of generated videos limits RIGVid…**\
> Advances in video generation have enabled substantial increases in video length, and RIGVid can stitch videos together by utilizing a high-level planner to execute tasks sequentially. Our experiments used Kling V1.6, which generated 5 sec videos. However, SOTA models now support much longer durations: Runway Gen-3 Alpha can generate up to 10 sec, Sora 2 Pro up to 25 sec, and Kling’s extension feature allows for videos up to 3 mins. These improvements show that video duration constraints are becoming less restrictive for RIGVid’s application. For tasks requiring longer horizons than current video generation allows, RIGVid can be readily integrated with high-level LLM planners that decompose tasks into sub-goals. To demonstrate this, we conducted preliminary experiments on the longer-horizon task: "place a fork and a spoon on a plate." We used GPT-o1 for sequential sub-task decomposition and Kling 2.1 for video generation. The final frame of each sub-task video serves as the initial frame for the next, with the same filtering mechanism (Sec. 3.1) applied throughout. We successfully generated complete two-step video sequences in 7 out of 10 trials, with initial single-step video generation succeeding in all 10 trials. One example of sequential generation is shown here [link1](https://drive.google.com/file/d/1KVDzZl4dHSipjos6T5uruDlBrUVWAUpF/view?usp=sharing) [link2](https://drive.google.com/file/d/1evYkzZCzE6nsGYiE9aiIkjRbGMaKiKbv/view?usp=sharing). These results demonstrate that RIGVid can successfully scale to longer-horizon tasks through hierarchical planning. Since our core contribution is reliable video-to-trajectory conversion, successfully generating and executing multi-step video plans shows a viable scaling path.
>
> **Generate robot gripper poses rather than hands…**\
> Analysis with current video generation models reveals that generating videos featuring robots significantly underperforms compared to human-centric videos, due to the predominance of human interaction data in model training. We experimented with generating videos featuring robot grippers and full robot embodiments. When we prompt to generate videos with xArm pouring water, the generated videos often omit the robot [link3](https://drive.google.com/file/d/1DwTwPENYe_WmzFP11GzoU_LEQQ45bgOi/view?usp=sharing) [link4](https://drive.google.com/file/d/1gAzbwQXULJN5A339_UbHjwEIF3vHUK-I/view?usp=sharing). We also prompted it to generate videos when the robot gripper is already in the starting frame. Again, the generated videos are implausible [link5](https://drive.google.com/file/d/1C2En8pviG63mf6wOrsQwabwVtbU2wD-U/view?usp=sharing). This limitation of video generators is, in fact, a strength for RIGVid, as our approach is embodiment-agnostic, as demonstrated by our successful transfer to ALOHA and bimanual setups (Sec. 4.5, Fig. 11) without requiring robot-specific training data. We will include a discussion about this in the revision.
>
> [1] Wen et al. BundleSDF, CVPR 2023\
> [2] Liu et al. Zero-1-to-3, ICCV 2023\
> [3] Huang et al. ReKep, CORL 2024\
> [4] Patel et al. IKER, ICRA 2025

---

> > ### Comment · Reviewer_Tugp · 2025-11-25
> >
> > Thanks to the authors for their concise and promising responses.
> >
> > Some of my earlier concerns remain partially unaddressed; it seems they may have to be left to future, more capable video‐generation models or to prior rule-based approaches (e.g., ReKep).
> > However, in my view, relying on a long video-generation phase before task execution is likely to remain a technique that works only on research paper.
> >
> > For the scope of the current submission and rebuttal, I believe there are no major issues. Given the training-free setting, the authors have already made full use of the available resources.
> > Accordingly, I will adjust my score to: marginally above the acceptance threshold.

---

### Meta-Review · Area_Chair_eacp · 2025-12-09

**Summary:**

Object models & deformables. Does RIGVid strictly require CAD/meshes; limits on non-rigid objects; why generate human hands vs robot grippers (Tugp)

Pre/post-grasp & recovery. How pre-grasp alignment/post-release are handled; whether there’s any explicit recovery after failures (Tugp)

Video length / long-horizon. Impact of short generations; need strategy for longer tasks (Tugp)

4D/3D reconstruction & tracking ablations. Try SOTA 4D recon (beyond mono-depth); test ICP as a tracker; broader tracking ablations (uft8)

Using robot data. Would finetuning video generators with robot data help (uft8)

Post-refinement. Any designs to refine trajectories/results after weak generations (uft8)

Presentation quality. Paper too dense; reorganize/clarify (bPbv)

Comparison to VLAs. Need conceptual and empirical context vs generalist VLA pipelines (bPbv)

Generalization. How well does it work without tuning/calibration; robustness to new scenes/embodiments (bPbv)

Statistical power & reporting. Add confidence intervals; expand qualitative failures (Jtgy)

Compute & latency. End-to-end wall-clock, attempts-per-success, throughput (Jtgy)

Why videos (not text/trajectories)? Show that VLM-on-video filtering beats text-only/trajectory-only filtering (Jtgy)

**Reviewer Concerns:**

CAD/meshes not strictly required: Addressed. Showed mesh-free tracking (BundleSDF) on pouring (noting current speed limits); option to synthesize meshes via image-to-3D; explained poor quality of robot-gripper video generations and argued for human-hand videos + embodiment-agnostic retargeting (Tugp)

Pre/post-grasp & recovery: Addressed. Detailed AnyGrasp-based pre-grasp, maintained closed-grip execution, recovery behavior and a VLM verification + re-generate path (Tugp)

Long-horizon: Addressed. Demonstrated hierarchical planning with stitched sub-videos (Tugp)

4D vs mono-depth; tracking ablations; ICP: Addressed. MonST3R swap ≈6× slower without success gains; 4D-DPM baseline underperforms; added ICP (lost tracking 6/10; much higher jitter) and broader tracking comparisons (uft8)

Robot-data finetuning: Addressed. Argued it likely helps but violates the paper’s zero-shot, demo-free scope (large data/compute requirements) (uft8)

Post-refinemen: Addressed. Current VLM filtering + smoothing; suggested policy-based refinement as future (but needs demos) (uft8)

Presentation: Addressed. Released a cleaner, rebalanced PDF per guidelines (bPbv)

VLA comparison: Addressed. Zero-shot pi0 comparison and conceptual positioning vs VLA pipelines (bPbv)

Generalization: Addressed. Zero-shot transfers (ALOHA, bimanual; diverse tasks) and note on single-frame depth calibration (bPbv)

Statistics: Addressed. Reported 95% CIs for task success (Jtgy)

Compute/latency: Addressed. Gave timings: video gen ~1 min, VLM filter <2 s, depth ~1 min, 6D tracking ~10 s; typically 1–2 attempts; total 2–5 min per trajectory (Jtgy)

Why videos: Addressed. New experiment: VLM-on-video filtering vs text-only LLM filtering of trajectories/captions; Pearson r ≈0.91 vs ≤0.12, resp; explained use of correlation and cited precedents (Jtgy)

Deformables: Partially addressed. Proposed keypoint-tracking extension and cited related work; not yet demonstrated at scale (Tugp)

Very long video-gen phase realism: Partially addressed. Reviewer still skeptical about practicality; authors show hierarchical planning but broader validation remains. (Tugp)

Heavier 4D pipelines: Partially addressed. Currently too slow for this setting; future faster 4D methods could be integrated (uft8)

**Reviewer Scores:**

reviewer Tugp increased his score from 4 to 6; I cannot guess which changes other reviewers could have been making
The other (original) scores where 4, 6, 6
That brings the overall picture to 4, 6, 6, 6

---

### Decision · Program_Chairs · 2026-01-26

Accept (Poster)